



# Description of historical and future projection simulations by the global coupled E3SMv1.0 model as used in CMIP6

Xue Zheng[1], Qing Li[2], Tian Zhou[3], Qi Tang[1], Luke P. Van Roekel[2], and Jean-Christophe Golaz[1]

[1]Cloud Processes Research and Modeling Group, Lawrence Livermore National Laboratory, Livermore, California, USA
[2]Fluid Dynamics and Solid Mechanics, Los Alamos National Laboratory, Los Alamos, New Mexico, USA
[3]Atmospheric Sciences and Global Change Division, Pacific Northwest National Laboratory, Richland, Washington, USA

**Correspondence:** Xue Zheng (zheng7@llnl.gov)





**Abstract.**

This paper documents the experimental setup and general features of the coupled historical and future climate simulations with the first version of the U.S. Department of Energy (DOE) Energy Exascale Earth System Model (E3SMv1.0). The future projected climate characteristics of E3SMv1.0 at the highest emission scenario (SSP5-8.5) designed in the Scenario Model Intercomparison Project (ScenarioMIP) and the SSP5-8.5 greenhouse gas (GHG) only forcing experiment are analyzed with a focus on regional responses of atmosphere, ocean, sea-ice, and land.

Due to its high climate sensitivity, E3SMv1.0 is one of the CMIP6 models with the largest surface warming by the end of the 21st century under the high-emission SSP5-8.5 scenario. The global mean precipitation change is highly correlated to the global temperature change, while the spatial pattern of the change in runoff responds to the precipitation changes. The oceanic mixed layer generally shoals throughout the global ocean. The sea ice, especially in the Northern Hemisphere, rapidly decreases with large seasonal variability. The annual mean AMOC is overly weak with a slower change relative to other CMIP6 models. We detect a significant polar amplification in E3SMv1.0 from the atmosphere, ocean, and sea ice.

Comparing the SSP5-8.5 all-forcing experiment with the GHG-only experiment, we find that the unmasking of the aerosol effects due to the decline of the aerosol loading in the future projection period causes accelerated warming in SSP5-8.5 all-forcing experiment. While the oceanic climate response is mainly controlled by the GHG forcing, the land runoff response is impacted primarily by forcings other than GHG over certain regions. However, the importance of the GHG forcing on the land runoff changes grows in the future climate projection period compared to the historical period.





# 1  Introduction

Compared to previous CMIP climate models, the latest CMIP phase 6 (CMIP6) models simulate a higher ensemble equi-
librium climate sensitivity (ECS) with a larger spread (Meehl et al., 2020; Zelinka et al., 2020). Within CMIP6, the Scenario
Model Intercomparison Project (ScenarioMIP) aims to generate multi-model climate projections for alternate scenarios of fu-
ture emissions and land-use changes produced with integrated assessment models. The climate model projections from the
ScenarioMIP experiments facilitate scientific understandings of future climate change. An ensemble analysis of the Scenario-
MIP participating global coupled Earth system models has shown that the global mean surface air temperature and surface
precipitation response of each individual model is highly correlated to its climate sensitivity, especially for the high-emission
scenario (Tebaldi et al., 2021).

The U.S. Department of Energy (DOE) Energy Exascale Earth System Model (E3SM) project is a new and ongoing climate
modeling effort to develop a state-of-the-art Earth system model. E3SM project aims to develop code optimized for DOE's
high-performance computing infrastructure and to advance Earth system prediction of changes in environmental variables that
are critical to energy-sector decisions, such as regional trends in air and water temperatures, water availability, storms and
heavy precipitation, coastal flooding, and sea-level rise (Bader et al., 2014; Leung et al., 2020). E3SM version 1.0 (E3SMv1.0)
at the standard horizontal resolution of $\sim 100\,km$ reported a high ECS of $5.3\,K$, a high transient climate response (TCR) of
$2.93\,K$, along with a strong aerosol-related effective radiative forcing ($ERF_{aero}$) of $-1.65\,W/m^2$ (Golaz et al., 2019). The
overly high ECS and strong $ERF_{aero}$ resulted in a delayed warming followed by an excessive warming trend during the
second half of the 20th century in the E3SMv1.0 historical ensemble (Golaz et al., 2019). It is expected that E3SMv1.0 will
be among the warmest models in terms of the global mean surface temperature in future climate projections due to its high
ECS and TCR. Through our participation in the ScenarioMIP project, we conducted future climate projection experiments
in a high-emission scenario with E3SMv1.0. Inspired by the Detection and Attribution MIP (DAMIP) project (Gillett et al.,
2016), we additionally conducted a set of historical and future projection simulations with GHG-only forcing for the high-
emission scenario to estimate the contribution of the greenhouse gas (GHG) emissions to observed global warming and the
future projected climate change in E3SMv1.0.

There are two main goals for this manuscript. Firstly, document future climate characteristics of E3SMv1.0 (which is a
model member of the ScenarioMIP project) at the highest emission scenario along with its historical climate evolution. In
particular, we describe regional responses of key climate components, namely atmosphere, ocean, sea-ice and land runoff
in the high-emission scenario simulated by E3SMv1.0. The currently available ScenarioMIP simulations from other CMIP6
models serve as a reference to characterize the E3SMv1.0 simulations. Secondly, we describe regional responses of key climate
components in the GHG-only simulations. The difference between the high-emission all-forcing experiment and the GHG-only
experiment is analyzed. Specifically, we compare the relative impacts of GHG forcing vs. other forcing on the different climate
components. In section 2, we present a brief model description of E3SMv1.0 and the detailed experimental setup. Section 3
includes all the results from these experiments, while the findings are summarized in section 4.



## 2 Model Description and Experiment Setup

### 2.1 E3SMv1.0 Model Description

Golaz et al. (2019) and references therein provide the full description of E3SMv1.0. Here, we only briefly describe the model information relevant to this study. E3SMv1.0 includes five Earth system components (atmosphere, ocean, sea ice, land, and rivers) and the coupler to interface these five components. The atmosphere component of E3SMv1.0, EAMv1, uses a spectral element dynamical core at 110-km resolution on a cubed sphere geometry. It has 72 layers with a top at approximately 60 km. The main atmosphere physics time step is 30 min. The ocean and sea ice components of E3SMv1.0 are developed based on the Model for Prediction Across Scales (MPAS) framework (Ringler et al., 2010): MPAS-Ocean and MPAS-Seaice. The MPAS framework uses Spherical Centroidal Voronoi Tessellations (SCVTs) for multi-resolution modeling. MPAS-Ocean and MPAS-Seaice share the same unstructured grid with a horizontal grid spacing varying from 30 km in the tropics and high-latitudes to 60 km in the mid-latitudes. The vertical discretization consists of 60 layers with thickness varying from 10 m at the surface to 250 m at depth. Ocean model time step is 10 min with a barotropic sub–time step of 40 s. The sea ice model time step is 30 min. The land component in E3SMv1.0, ELMv1, is developed from the Community Land Model version 4.5. The time step of ELMv1 is 30 min. The river runoff component of E3SMv1.0 is the Model for Scale Adaptive River Transport (MOSART). In the standard E3SMv1.0 1-degree resolution, MOSART uses a regular latitude-longitude grid with the resolution of $0.5°$. The time step of MOSART is 1 hour. The component coupler in E3SMv1.0 is the Common Infrastructure for Modeling the Earth. The coupling frequency for all components is 30 min except for MOSART which communicates every 3 hours.

The key source code git hash numbers involved in the E3SMv1.0 simulation campaign are domcumented in Golaz et al. (2019). A maintenance branch (maint-1.0; https://github.com/E3SM-Project/E3SM/tree/maint-1.0) has also been created and maintained to reproduce these E3SMv1.0 simulations performed for this study. The run scripts used to set up simulations beyond the default configuration in the model compset and submit jobs for these experiments are also archived to reproduce these simulations (see *Code availability*).

### 2.2 The CMIP6 historical experiment

The E3SMv1.0 CMIP6 historical simulations follow the CMIP6 Diagnosis, Evaluation, and Characterization of Klima (DECK) specifications (Eyring et al., 2016). Golaz et al. (2019) documents the model input data (i.e., input4MIPS data), the spin-up, initialization and the tuning efforts for the E3SMv1.0 preindustrial control simulation ($piControl$). Five ensemble members of the CMIP6 historical simulations ($historical\_Hn$) were initialized from 1 January of five different years of the $piControl$ simulation (Golaz et al., 2019, Table 2). These $historical\_Hn$ simulations cover the 1850-2014 period.

### 2.3 ScenarioMIP SSP5-8.5 Experiment

The E3SMv1.0 future climate projections adopt the Shared Socio-Economic Pathway-Representative Concentration Pathway (SSP-RCP) framework of the ScenarioMIP experiments (O'Neill et al., 2016). The ScenarioMIP experimental design





includes a set of eight pathways of future emissions, concentrations and land use, with additional ensemble members and long-term extensions to facilitate future research on mitigation, adaptation and residual climate impacts (O'Neill et al., 2016). We conducted the future climate projection experiment with the high-emission SSP5-8.5 scenario, which represents the upper

end of the scenarios in terms of fossil fuel use, food demand, energy use and greenhouse gas emissions (Kriegler et al., 2017). The high-level results from these future projection runs were included in the CMIP6 ScenarioMIP paper (Tebaldi et al., 2021). The SSP5-8.5 Scenario experiment produces a radiative forcing of 8.5 W m$^{-2}$ in the year of 2100 (O'Neill et al., 2016). The relatively high forcing level reached by this scenario enables the model to simulate the potential responses of the Earth system components over a range of global average radiative forcing and temperature changes that are larger than in lower emission

scenarios by the end of 21th century.

Kriegler et al. (2017) describe forcings including global spatial distributions of emissions and concentrations of greenhouse gases, ozone concentrations, aerosols, land use, and other natural forcings, in particular solar forcing and volcanic emissions, for the ScenarioMIP SSP5-8.5 experiment. To keep the consistency through the harmonization of emissions, concentrations, and land use across scenarios and between the SSP5-8.5 simulations and historical simulations, five ensemble members of

the ScenarioMIP simulations ($future\_Pn$-SSP5-8.5) use the conditions at the end of the $historical\_Hn$ simulations (31 December 2014) as the initial conditions for future climate projections. The E3SM future climate simulations and the CMIP6 historical simulations are performed with the same model configuration, including the input data processing for GHGs and aerosols emissions (Golaz et al., 2019). We will link these two experiments and analyze the present-day climate and the future climate projections together to see if these are any disruptions at the first years of the SSP5-8.5 simulations related to the

historical simulations.

## 2.4   GHG-only Experiment

As one of the CMIP6-Endorsed MIPs, Detection and Attribution MIP (DAMIP) aims to estimate the contributions of anthropogenic and natural forcing changes to observed global warming as well as to observed global and regional changes in other climate variables (Gillett et al., 2016). While there are a number of experimental designs covering the historical period and

the future projection period with the SSP2-4.5 future scenario in DAMIP, we simply adopt the "only" approach for the GHG forcing in the historical period and the SSP5-8.5 future scenario. We conducted a total of three ensemble members of the GHG-only simulations for the historical period and the future projection period, respectively. The model setting of these GHG-only historical simulations ($historical\_Hn$-GHG) are the same as the $historical\_Hn$ runs except that all forcings other than GHG forcing are held at preindustrial values. The GHG-only future projection simulations ($future\_Pn$-SSP5-8.5-GHG) use the

end of the $historical\_Hn$ simulations as the starting points and use the GHG forcing in the SSP5-8.5 scenario, while the other forcings are still set at the same preindustrial values as $historical\_Hn$-GHG. Similarly to the CMIP6 historical experiment and the ScenarioMIP SSP5-8.5 experiment, we connect the GHG-only historical and future projection experiments together to analyze the climate responses from year 1850 to year 2099.





## 3  Results

### 3.1  Historical simulation and SSP5-8.5 experiment

#### 3.1.1  Atmosphere Climatology

Before we analyze the global mean or zonal mean of the variables, the monthly variables are regridded to $1°$ lat-lon grids with the first-order conservative remapping method through netCDF Operators (NCO) version 4.8.1. CMIP6 models project an overall higher warming with a larger intermodel spread for different forcing levels, particularly for the high emission SSP5-8.5 scenario, compared to the corresponded CMIP5 future climate projects. These changes are likely due to the different experimental designs between CMIP5 and CMIP6 experiments and the higher climate sensitivities in a subset of the CMIP6 models (Tebaldi et al., 2021). To adopt the CMIP6 models as the reference, this study analyzes these CMIP6 model runs of which both the historical simulation and SSP5-8.5 simulation are available at the time of writing. Only one ensemble member (r1i1p1f1) of the CMIP6 model runs are included in this study.

Figure 1 presents the time evolution of annual global mean near-surface air temperature ($T_{air}$) anomalies and surface precipitation rate anomalies with respect to 1850-1869 from E3SMv1.0 along with the other CMIP6 models. As shown in Golaz et al. (2019), E3SMv1.0 simulated global mean $T_{air}$ anomalies during the historical period demonstrates a prolonged cooling after 1950 and then a rapid warming around 2000. E3SMv1.0 is at the lower end of the model range during the prolonged cooling period (Fig. 1a). However, the $T_{air}$ anomalies in E3SMv1.0 catch up and reach the middle range of the CMIP6 model spread by the end of the historical run (year 2014) due to the rapid warming. This rapid warming in E3SMv1.0 continues in the SSP5-8.5 experiment at a speed faster than most of CMIP6 models. Near the end of the 21st century, E3SMv1.0 projects one of the warmest $T_{air}$ anomalies at $\sim 8$ K, consistent with the overly strong TCR and ECS. The global mean precipitation rate is driven by the energy balance between the radiative cooling and the latent heating (O'Gorman et al., 2012); the change in precipitation is mainly set by the Clausius–Clapeyron relation as an approximately 7% $K^{-1}$ increase in atmospheric water vapour content (e.g. Stephens and Hu, 2010). As a result, the time evolution of the global mean surface precipitation rate (Fig. 1b) is strongly correlated to the $T_{air}$ trend with larger inter-annual fluctuations. The global mean precipitation in E3SMv1.0 increases by >0.3 mm day$^{-1}$ from the end of the historical period (year 2014) to the end of the future projection (year 2099). The spread of the E3SMv1.0 ensemble members is much smaller than the CMIP6 intermodel spread throughout the historical and future climate projection periods for both $T_{air}$ and surface precipitation rate anomalies.

The global distribution of the $T_{air}$ and surface precipitation change in E3SMv1.0 resembles the ensemble average pattern from the ScenarioMIP participant models for the SSP5-8.5 scenario (Tebaldi et al., 2021) based on the global maps of $T_{air}$ (Fig. 2) and surface precipitation rate (Fig. 3) for year 2070-2099 from the SSP5-8.5 simulations and these for year 1985-2014 from the historical simulations. The global mean precipitation increases by $\sim 10\%$ due to the warmer climate. In the tropics, the rain bands over the intertropical convergence zone (ITCZ) are all strengthened, while the precipitation in the Amazon and Central America is reduced. In the mid- and high-latitudes, the precipitation over both the North America and the Eurasian continents increases in the SSP5-8.5 simulations except the Mediterranean region. The major drying regions include the Mediterranean





region, Central America, the Amazon region, southern Africa, and western Australia. Note that these drying regions also show dry biases in the historical runs based on the Global Precipitation Climatology Project v2.2 observational estimate for year 1979-2014 (Golaz et al., 2019). The magnitude of the mid-latitude continental summer warm biases in the present climate in

CMIP5 models were found to be closely linked to the projected climate change amplification in the local warming (Cheruy et al., 2014). Due to the strong land-atmosphere coupling, we speculate that the magnitude of the precipitation bias in the current climate simulation also links to the projected climate change amplification in the drying signal, which needs further multi-model investigation to confirm in the future study.

Treating the available CMIP6 models as a reference, we sort the global mean $T_{air}$ changes between year 2070-2099 and

year 1850-1869 in the SSP5-8.5 simulations from the available CMIP6 models (blue bars) and E3SMv1.0 ensemble members (red bars) from the lowest warming to the highest warming (Fig. 4). The global mean $T_{air}$ changes in the E3SMv1.0 ensemble members are comparable to the CMIP6 models with the highest warming. Besides the global mean $T_{air}$ anomaly and surface precipitation rate, we analyze the time evolution of zonal mean $T_{air}$ and shortwave cloud radiative effect (SWCRE) to characterize the regional temperature changes and cloud responses simulated by E3SMv1.0 in the historical and future climate

simulations. We also analyze the CMIP6 models with low ($0 - 20th$ percentiles) and median warming ($40 - 60th$ percentiles) to better illustrate the model inter-comparison of these regional patterns. Golaz et al. (2019) compared the time evolution of the sea surface temperature (SST) anomalies in the Northern and Southern Hemispheres with observational records and found that SST in the Northern Hemisphere is the main cause of the global mean bias in the E3SMv1.0 simulated $T_{air}$ anomalies during the historical period. Figure 5a shows the time evolution of the zonal mean $T_{air}$ anomalies from all E3SMv1.0 ensemble

members, revealing a rapid warming in the Arctic and a clear warming asymmetry between the Northern Hemisphere and the Southern Hemisphere in the SSP5-8.5 simulations. To better detect the regional pattern in $T_{air}$ from the historical simulations and the future climate simulations, we calculated the local change of $T_{air}$ anomalies (Fig. 5b) by subtracting the zonal mean $T_{air}$ anomalies for year 1850-1869 from the time evolution of the zonal mean $T_{air}$ anomalies (i.e. Fig. 5a). The local change of $T_{air}$ anomalies reveals a continuous enhanced cooling in the Northern Hemisphere (10°N-60°N) lasting from 1870 to 2000

in E3SMv1.0 (Fig. 5b), which is the main contributor to the prolonged cooling shown in the global mean $T_{air}$ anomaly (Fig. 1a). The time evolution of SWCRE (Fig. 6) indicates that this continuous enhanced cooling in the Northern Hemisphere before 2000 corresponds clearly to an enhanced negative SWCRE over the same region. The local changes in $T_{air}$ and SWCRE from CMIP6 models with low and median warming have no signal of such a continuous cooling in the Northern Hemisphere in the historical simulations (Fig. 6a-b).

After year 2000, $T_{air}$ in E3SMv1.0 and other CMIP6 models starts increasing, especially over the polar regions. E3SMv1.0 shows a clearly faster warming over the Arctics than CMIP6 models with low and median warming, indicating a stronger polar amplification in E3SMv1.0. The stronger polar amplification tends to be associated with lower sea-ice concentrations, the weaker poleward ocean heat transport at high latitudes, and increases in cloud cover over the polar regions (Holland and Bitz, 2003; Pithan and Mauritsen, 2014; Cohen et al., 2020). Indeed, the negative changes in SWCRE over the polar regions,

an indicator of the increased polar cloud amount, in E3SMv1.0 are stronger and enhance faster than CMIP6 models with low and median warming after year 2000 (Fig. 6). However, the regions with a strong negative SWCRE changes in E3SMv1.0





are confined to higher latitudes in both hemispheres. Especially after year 2050, the region with a strong negative SWCRE changes in the Northern Hemisphere retreats to latitudes higher than 60°N in E3SMv1.0, while the weakening of the negative SWCRE (i.e. a positive change in SWCRE in Fig. 6) becomes much stronger and broader in low- and mid-latitudes compared

with the CMIP6 models with low and median warming. Overall, the clouds in E3SMv1.0 show a slightly stronger but more confined negative SW feedback in the high latitudes, while a much stronger and broader positive SW feedback in the low- and mid-latitudes relative to CMIP6 models with low and median warming. The difference in SWCRE between E3SMv1.0 and CMIP6 models with low and median warming becomes substantial after 2050 (Fig. 6). Meanwhile, the near surface warming across the low-, mid- and high- latitudes after 2050 (Fig. 5) indicates that the strong warming in E3SMv1.0 is primarily due

to a stronger polar amplification and stronger positive cloud feedbacks from decreasing extratropical low cloud coverage and albedo (Zelinka et al., 2020). Throughout the historical period and the future climate projection period, E3SMv1.0 produces an inter-hemispheric asymmetric cooling and then an inter-hemispheric asymmetric warming, both of which are closely linked to the cloud responses, especially in the Northern Hemisphere. A recent study (Wang et al., 2021) found that CMIP6 models with a more positive cloud feedback tend to have a stronger cooling effect from aerosol-cloud interactions. The CMIP6 models with

a weak aerosol indirect effect and a low cloud feedback are more consistent with the observed warming asymmetry during the mid to late 20th century. We will discuss the impact of the strong aerosol indirect effect in E3SMv1.0 on the future climate projection in Section 3.2.1.

### 3.1.2 Ocean and Sea-ice

The time evolution of SST (Fig. 7) in the SSP5-8.5 simulations is consistent with that of $T_{air}$ in Fig. 1a, with a much faster

warming than most of other CMIP6 models. By the end of year 2099, SST in E3SMv1.0 increases by $\sim 5°C$, the strongest warming among the CMIP6 models. Note that the CMIP6 models used in section 3.1.1 are slightly different from the CMIP6 models in this section due to the availability of model output varying between the atmospheric variables, ocean and sea-ice variables. Fig. 8 shows the changes in the ensemble averaged SST between the period of 2070-2099 and 1985-2014 in the boreal winter (Jan., Feb., Mar.) and summer (Jul., Aug., Sep.). We see a warming in excess of 2 °C almost everywhere in the

global ocean, especially in the high latitudes in boreal summer, when the changes in SST can reach over 10 °C locally. This is consistent with the strong polar amplification described in section 3.1.1. Correspondingly, there is a strong freshening in the Arctic in both seasons as illustrated by the changes in the sea surface salinity (SSS) in Fig. 9. This is a result of the melting sea ice in the Arctic due to the polar amplification of global warming. The overall decrease in SSS in the North Atlantic and increase in SST in the South Atlantic may be related to the weakening of the Atlantic meridional overturning circulation (AMOC; See

Fig. 11 and the corresponding discussion). The mixed layer generally shoals due to an overall warming throughout the global ocean. This is especially true for the winter mixed layer depth, e.g., the Kuroshio and Gulf Stream extensions in boreal winter and the Southern Ocean in boreal summer in Fig. 10.

Fig. 11 compares the simulated annual mean AMOC in E3SMv1.0 and other CMIP6 models, measured by the maximum streamfunction nearest to the RAPID Array latitude 26°N. The mean AMOC simulated in the E3SMv1.0 historical ensemble

is weaker than the observed mean (Golaz et al., 2019). Here we see that it is also at the lower end of the CMIP6 ensemble





for the future climate projection. Possible reasons have been discussed in Golaz et al. (2019), including the spurious diapycnal mixing, poor representation of the Nordic overflows and critical passageways transporting freshwater from the Arctic, as well as excess simulated sea ice in the Labrador Sea in E3SMv1.0.

Interestingly, E3SMv1.0 also exhibits the slowest weakening of AMOC among all the CMIP6 models available in the SSP5-8.5 experiment. This is better seen in Fig. 12, which shows the changes in AMOC and SST versus the mean AMOC and SST at the beginning of the SSP5-8.5 simulations and the relation between the warming SST and weakening AMOC for the ensemble of CMIP6 models. Consistent with Hu et al. (2020), a weaker simulated AMOC is often associated with a weaker change in AMOC in response to the SSP5-8.5 forcing (left panel), and often corresponds to a faster warming (right panel). And this relation seems to be valid for the majority of the CMIP6 models in addition to E3SMv1.0 and CESM2 explored in their study

Hu et al. (2020).

The time series of total sea ice area in March and September for E3SMv1.0 and CMIP6 models show that the Northern Hemispheric sea ice reduction is faster than most of CMIP6 models with large seasonal variability. While it is comparable with other CMIP6 models in March at the beginning of the SSP5-8.5 simulations (Fig. 13a), the sea ice in September is less than most of CMIP6 models and rapidly deceases to zero after year 2040 (Fig. 13b). The Northern Hemispheric sea ice in March

rapidly decreases around 2050 and reduces to near zero after year 2080. The Southern Hemispheric (SH) sea ice reduction is within the wide model spread for both March and September (Fig. 13c-d). Analyses conducted during the development of E3SMv2.0 have shown that an accounting error in the exchange of frazil ice mass between the ocean and the atmosphere largely contributes this strong reduction.

### 3.1.3 Land climatology

Runoff is one of the most representative variables to reflect the land climatology. The spatial pattern of the mean annual runoff for year 1985-2014 and year 2070-2099 (Fig. 14 a and b) are generally similar. The patterns are consistent with the DECK simulation results by E3SMv1.0, which had noticeable wet biases over the arid regions such as Australia and western United States and dry biases over the northern South America (Golaz et al., 2019). The runoff change in the SSP5-8.5 simulations (Fig. 14 c) agrees with previous climate change studies (e.g. Nohara et al., 2006) and other CMIP6 model predictions (e.g. Cook

et al., 2020) with decreased runoff in the Mediterranean region, southern Africa, southern North America, Central America, Australia, and increased runoff in high latitudes of the Northern Hemisphere, central Africa, as well as southern to eastern Asia. Similar to other land surface models, annual runoff predicted by ELMv1 is highly correlated to the precipitation changes (Fig. 3). Given that the spatial distribution and bias of the runoff are highly consistent with those in precipitation for E3SMv1.0, it is fair to presume that the position of E3SMv1.0 runoff for the SSP5-8.5 future scenario is similar to the position for precipitation

among other CMIP6 models.

### 3.2  GHG-only Experiment

As shown in Fig. 1 and Fig. 2, the climate in the GHG-only experiment warms more rapidly than the all-forcing experiment in the historical simulations and it is warmer than the SSP5-8.5 simulations. Meanwhile, the SST is warmer, the sea-ice amount





is less and the AMOC is further weakened in the GHG-only experiment compared to the SSP5-8.5 experiment (Fig. 7, Fig. 13,
and Fig. 11). Previous modeling and observational studies have controversial findings on the contribution of GHG forcing vs.
other forcings to the historical climate change and the future climate projection. While studies indicate that human–induced
GHG forcing dominate observed global warming since the mid–20th century (Jones et al., 2013), some studies found that both
GHG and aerosol changes contribute warming during the 21st century (Gillett et al., 2012) and aerosol forcing is found to
determine intermodel variations in the historical surface temperature for CMIP5 models (Rotstayn et al., 2015). Beyond these
high-level features, the following subsections focus on the difference between the historical and SSP5-8.5 experiments (i.e., the
all-forcing simulations) and the corresponding GHG-only experiments, which will shed light on the contribution of the GHG
forcing to the E3SMv1.0 simulated climate change in the history and future projection relative to the other forcings.

### 3.2.1   Atmospheric Responses

In the absence of the anthropogenic aerosol, the global mean $T_{air}$ increases monotonically in the GHG-only historical
experiment (Fig. 1). Unlike in the all-forcing historical experiment, the $T_{air}$ and SWCRE changes show no signal of cooling
and enhanced cooling effect in the Northern Hemisphere over the regions with a clearly higher aerosol load in the all-forcing
historical experiment, a significant portion of which is sulfate aerosol (Fig. 15a,b,e,f). The net CRE shows an extra strong
cooling effect up to -5 W/m2 over the previously mentioned region/period in the all-forcing historical experiment (Fig. 15d).
This further confirms that the overly strong $ERF_{aero}$, including the strong aerosol-cloud interactions, causes the prolonged
cooling between 1960 and 2000 and the delayed warming after 2000 in the E3SMv1.0 historical experiment. Near the end of
the historical simulations, $T_{air}$ is warmer almost everywhere in the GHG-only historical experiment, especially over the Arctic
(Fig. 2, mid-panel, Fig. 5b vs. Fig. 15a).

In the SSP5-8.5 GHG-only experiment, although $T_{air}$ increases with a spatial pattern similar to the SSP5-8.5 all-forcing
experiment, the warming slows down compared to the SSP5-8.5 all-forcing experiment based on the zonal mean $T_{air}$ trend
over the last 50 years of the simulations. In particular, the polar amplification is weaker in the GHG-only experiment (Fig.
15c). The difference in the net CRE between the all-forcing experiment and the GHG-only experiment during the future
climate project period contributes to the slowing down of the warming (Fig. 15c), implying an additional warming component
from the unmasking of the aerosol effects in the all-forcing experiment likely linked to the decline of the aerosol load in the
future climate projection period (Fig. 15e,f). The global distribution of the warming between year 2070-2099 and year 1985-
2014 from the GHG-only experiment shows weaker warming in the Northern Hemisphere than the all-forcing experiment with
a polar-ward increased difference (Fig. 2 bottom). Previous studies detected rapid near-term warming through the 21st century
driven by decrease in aerosols in CMIP5 models (e.g., Chalmers et al., 2012; Levy II et al., 2013), whereas aerosol emission
reduction caused gradual warming in other CMIP5 studies (e.g., Gillett and Salzen, 2013). Further simulations and analyses
will be needed to fully understand the mechanism underlining the accelerated warming in the Northern Hemisphere, which is
beyond the GHG-induced warming.

During the historical period, the global precipitation in the GHG-only simulations is larger than that in the all-forcing
simulations by 0.13 mm day$^{-1}$ ($\sim 4\%$) at the end of the historical period (Fig. 1). The global maps (middle panel of Fig. 3)




indicate that the GHG-only experiment is overall similar to the all-forcing experiment at the end of the historical period with changes resembling the SSP5-8.5 experiment except for the smaller magnitude, i.e. slightly increased precipitation in the tropics
and the mid-lat oceans. The changes in precipitation rate between the future climate projection and the historical period from the all-forcing experiment and GHG-only experiment (bottom panel of Fig. 3) suggest that the magnitude of the precipitation change is larger in the all-forcing experiment especially for the drying regions, e.g. the central and eastern south Pacific, the south Indian ocean, Australia and the Amazon region. Meanwhile, the changes over regions with increased precipitation in the future climate projects are generally larger in the all-forcing experiment than that in the GHG-only experiment, e.g. the North
Indian ocean, the India peninsular and the Tibetan Plateau, the mid-latitude Eurasia, and the coastal regions of the Northern Pacific. One exception is that the drying signal along the tropical eastern Pacific to Central America is stronger in the GHG-only experiment.

### 3.2.2 Ocean and Sea-ice Responses

Comparing the right panels in Figure 8-9 to the left panels, we find that changes in SST, SSS, mixed layer depth from
the SSP5-8.5 all-forcing experiment vs. the historical experiment are mostly similar to these from the SSP5-8.5 GHG-only experiment vs. the historical GHG-only experiment, suggesting that GHG emissions dominate the changes in oceanic mean climate relative to other forcings. One exception is that SSS over the subtropical South Pacific increases more significantly in the all-forcing experiment for both boreal winter and summer (Fig. 9). Given that the mixed layer depths (Fig. 10) and SST (Fig. 8) are unchanged it is likely the changes in SSS are driven by atmospheric forcing, which is indeed evident in the bottom
panel of Fig. 3 that the projected reduction of precipitation in the all-forcing experiment is stronger than that in the GHG-only experiment over this region. We also note that the strength of the AMOC is slightly weaker in the GHG-only experiment during most of the period (Fig 11). This could be due to the increase in $T_{air}$ (Fig. 1), which would decrease North Atlantic surface density, thus, reducing deep convection and AMOC. The SSP5-8.5 all-forcing and GHG-only experiments converge toward the end of the simulations. Given that the AMOC is so weak in both experiments, this is likely nearing an "off" state.
The sea ice extent for the SSP5-8.5 all-forcing experiment and the SSP5-8.5 GHG-only experiment are shown in the black and gray lines respectively in Fig. 13. Overall the trends are similar, which is not surprising given the strong warming in both experiments. However, we do note that the sea ice extent during boreal winter in the GHG-only experiment begins lower, but decreases at the same rate as the all-forcing simulation until approximately 2050, when the GHG-only simulation decreases rapidly. The sea ice extent in the all-forcing experiment remains larger for another decade before dropping rapidly. Given the
small changes in SST (Fig. 8) between the all-forcing experiment and the GHG-only experiment, it is unlikely that ocean dynamics drive the enhanced sea ice loss in the GHG-only experiment. Further, the difference in the net CRE between the all-forcing experiment and the GHG-only experiment (Fig. 15) over the polar regions suggests more warming from cloud radiative effect for the all-forcing experiment, which is counter to what is observed in Fig. 13. Instead, the further decrease in sea ice in the GHG-only experiment is likely driven by the increased $T_{air}$ in the GHG-only experiments during the future
climate projection period (e.g. Fig. 1), because the SSP5-8.5 GHG-only experiment starts at a warmer state from the historical GHG-only experiment, and the sea ice is correspondingly starting from a decreased extent.





### 3.2.3 Land Responses

The land responses were first examined by comparing the runoff distributions driven by all forcing and GHG-only forcing (Fig. 14). No obvious differences can be identified between the runoff during either historical or future period. However,
noticeable deviation can be seen in the historical to future changes (i.e. between Fig. 14 c and f). Specifically, in southern North America, southern Africa, and eastern Asia, GHG-only forcing leads to a greater decline in future runoff than all forcing condition, while in central Africa, the GHG-only forcing tends to have less runoff increase in the future than the all forcing condition. To further examine the time evolution of runoff responses in the SSP5-8.5 all-forcing experiment and the SSP5-8.5 GHG-only experiment, we applied the two-sample Kolmogorov-Smirnov (K-S) test on the annual mean runoff at gridcell level.
The two-sample K–S test has been widely applied in climate studies (e.g. Wang et al., 2008; Zhou et al., 2018; Gaetani et al., 2020) to examine whether two data samples are from a same distribution by comparing their empirical cumulative distributions functions (eCDFs). The null hypothesis ($H0$) is that the values of two data sets are from the same continuous distribution, which can be rejected at a significant level $\alpha$ if

$$D_{m,n} > i(\alpha)\sqrt{\frac{m+n}{m \times n}}$$

where $m, n$ are the sample sizes of the two samples; $D_{m,n}$ is the distance between the eCDFs of the two samples; $i(\alpha)$ is the inverse of the Kolmogorov distribution at $\alpha$. The smaller $D_{m,n}$, the more similarity between the to eCDFs. In this study, the K-S test was conducted for mean annual runoff at every land grid cell between the SSP5-8.5 all-forcing experiment and and GHG-only experiment with $\alpha = 0.01$. Similar to previous sections for other climate components, we selected two 30-year periods to represent historical (1985-2014) and future (2070-2099) conditions. In the following context we will use letter $R$ to
indicate $H0$ was rejected, meaning the two samples (i.e. runoff from the all-forcing experiment and GHG-only experiment) are from different distributions; and letter $F$ to indicate $H0$ was fail to reject, meaning the two samples are from the same distribution. Two representative pixels were picked to further demonstrate the changes in runoff time series as well as the eCDFs for the two different directions of changing (Fig. A1 for $F$ to $R$, and Fig. A2 for $R$ to $F$).

The global distribution of the K-S test results for the historical period (Fig. 16 a) were $F$ in most areas but $R$ in Greenland,
Australia, central and northwest of North America, and eastern Asia. For the future climate projection period (Fig. 16 b), the $F$ area was generally expanded, indicating the difference in local runoff enlarges in the future climate projection experiments. Note that, some areas turn from $R$ to $F$, such as Greenland, Australia, and Alaska, meaning that the time evolution of the local runoff from the all-forcing experiment and GHG-only experiment become closer to each other during the future climate projection period. The difference in local runoff during both time periods will contribute to the difference in the projected
runoff change between the SSP5-8.5 all-forcing experiment and the GHG-only experiment (Fig. 14 c and f). Therefore, the regions with $F$ in either period (Fig. 16 a and b) tend to have notable difference in the projected runoff changes shown between the bottom left panel and the bottom right panel of Figure 14.

The difference between the K-S test results (Fig. 16 c) clearly shows the regions with a switch in runoff changes from the historical period to the future climate projection period. Areas where K-S test results changed from $R$ to $F$ (i.e. purple regions
in Fig. 16 c). We also noticed that the K-S test results in some areas, such as central Africa, changed from $F$ to $R$ which is





opposite to the general trend. Overall, about 14% of the global area changed the results from F to R (orange color) and 26% area changed from R to F (purple color), nearly doubled. This suggests that the runoff distributions from the SSP5-8.5 all-forcing experiment and GHG-only experiment tend to be similar in general during the future period than during the historical period, implying that the GHG emission plays a dominant role among all forcings in terms of runoff for the future climate projection

period. Instead of investigating changes in annual mean (Fig. 14), K-S tests focus on the distributional changes and thus provide additional information associate with systematic alteration in the time series. One limitation of our current annual scale runoff analysis is that it did not fully address the snow dynamic changes, which mostly associate with seasonal shifts in runoff due to changes in snow accumulation and melting processes as well as snow versus rain partitioning in the total precipitation (Cook et al., 2020; Knutti and Sedláček, 2013). Therefore, to better understand the contribution of forcings on the runoff changes,

seasonal runoff analysis will be needed in the future studies.

## 4    Summary and conclusions

In this paper, we describe the experimental setup and general features of the coupled historical and future projection simulations that E3SMv1.0 contributes to ScenarioMIP of CMIP6. We conducted two sets of coupled E3SMv1.0 experiments in the highest-emission SSP5-8.5 scenario: 1) the all-forcing experiment designed in the ScenarioMIP project, and 2) the SSP5-8.5

GHG-only experiment inspired by the DAMIP project. Both experiments include the historical simulations (years 1850-2014) and the future projection simulations (years 2015-2099). Five ensemble members were generated for the SSP5-8.5 all-forcing experiments, while three ensemble members were conducted for the GHG-only experiments. Analyzing the ensemble means, we describe the global and regional responses of atmosphere, ocean, sea-ice and land runoff during the whole period. The currently available ScenarioMIP simulations from other CMIP6 models serve as a reference to characterize the E3SMv1.0

simulations. Furthermore, we estimate regional responses of key climate components in the GHG-only simulations in comparison with these in the all-forcing experiment. The relative impacts of GHG forcing vs. other forcing on the future projections of the different climate components are analyzed and reveal the following features about the future climate projection by E3SMv1.0:

1. E3SMv1.0 is one of the CMIP6 models with the largest surface warming by the end of the 21st century under the SP5-

375        8.5 scenario, which is consistent with the overly strong TCR and ECS of E3SMv1.0. The global surface precipitation rate increases along with the surface warming. The regional patterns of the projected precipitation change by E3SMv1.0 are consistent with the ensemble average pattern from the ScenarioMIP participant models for the SSP5-8.5 scenario. The regions with significantly increased precipitation include ITCZ, North America, and most of the Eurasian continent. The major drying regions include the Mediterranean region, Central America, the Amazon region, southern Africa, and

380        western Australia, regions where dry biases exist in the historical simulations (Golaz et al., 2019).The spatial pattern of the change in land runoff is highly correlated to the precipitation changes.

2. The global SST increase is similar to $T_{air}$ with a much faster warming than most of other CMIP6 models. Meanwhile, the oceanic mixed layer generally shoals due to the overall warming throughout the global ocean. The sea ice reduction,





especially over the Northern Hemisphere, is faster than most of the CMIP6 models with large seasonal variability. The annual mean AMOC is at the lower end of the CMIP6 ensemble for the future climate projection. The change in AMOC is weaker in response to the SSP5-8.5 forcing, which likely contributes to the faster warming in E3SMv1.0.

3. There is a strong signal of polar amplification in E3SMv1.0 shown as a strong $T_{air}$ and SST warming in the Arctic. It is associated with a weaker AMOC, lower sea ice concentration and faster sea ice melt, reduced SSS, and a increased clouds over the Arctic.

4. The time evolution of the zonal mean $T_{air}$ shows that E3SMv1.0 has a strong cooling in the Northern Hemispheric mid-latitudes between year 1900 and 2000, which is consistent with the peak aerosol optical depth, supporting the hypothesis of an overly strong aerosol indirect effect.

5. In the SSP5-8.5 GHG-only experiment, the global mean $T_{air}$ is higher than the SSP5-8.5 all-forcing experiment in both historical and future projection periods. The warming, however, slows down with a weaker polar amplification compared to the all-forcing experiment over the last 50 years of the future projection simulations. The accelerated warming shown in the SSP5-8.5 all-forcing experiment exceeds the GHG-induced warming. The accelerated warming is likely linked to the unmasking of the aerosol effects from the decline of the aerosol loading in the future projection period.

6. Comparing the SSP5-8.5 experiment with the GHG-only experiment suggests that the GHG forcing dominates the control of the oceanic climate change. In contrast, land runoff analyses found that the runoff change between the SSP5-8.5 all-forcing experiment and GHG-only experiment is larger over certain regions, e.g. southern North America, southern Africa, central Africa, and eastern Asia especially during the historical period. But the runoff distributions from the all-forcing experiment and the GHG-only experiment tend to become more similar during the future period as the impact of GHG forcing grows and becomes dominant.

As discussed in section 3, this paper mainly describes the experiments and present the most notable features revealed in these experiments. Further model sensitivity tests and in-depth model diagnostics combined with observational references will be required to fully understand the mechanisms causing these general features documented in this study.

**Appendix A**

Table A1 lists the CMIP6 models, of which the historical experiment and the SSP5-8.5 all-forcing experiment were included in this study. All those model data have been released the Earth System Grid Federation. The DOIs for the data and the reference for the model are listed in the table as well.

Figure A1 and Figure A2 demonstrate how the two-sample K-S test determines the changing directions of the runoff change based on the time serier of runoff over a grid point. Figure A1 shows that the runoff change switches from the same distribution during 1985-2014 to the different distribution during 2070-2099 at 9.5°S, 40.5°W (red circle in Figure 16c). Figure A2 shows





that the runoff change switches from the different distribution during 1985-2014 to the same distribution during 2070-2099 at
22.5°N, 100.5°E (green circle in Figure 16c).

*Data availability.*  The E3SMv1.0 model code is released at https://doi.org/10.11578/E3SM/dc.20180418.36.

The E3SMv1.0 historical simulations and future climate simulations data can be accessed on the Earth System Grid Federation (ESGF)
platform (https://esgf-node.llnl.gov/search/e3sm/).

https://doi.org/10.22033/ESGF/CMIP6.4497.

The run scripts used to set up simulations in this study are available online (https://doi.org/10.5281/zenodo.5498235).

*Author contributions.*  XZ coordinated the writing of this article. JG and QT coordinated, conducted and archived the simulations for this
study. XZ and JG analyzed and documented the atmospheric results. QL analyzed the ocean and sea-ice results from E3SMv1.0 and other
CMIP6 models. QL and LVR wrote the sections on the ocean and sea-ice. TZ analyzed the land runoff from these E3SMv1.0 simulations
and wrote the related sections. All co-authors gave feedback on the manuscript.

*Competing interests.*  The authors declare that they have no conflict of interest.

*Acknowledgements.*  This research was supported by E3SM project funded by the Office of Biological and Environmental Research in the
U.S. Department of Energy's Office of Science. This research used resources of the National Energy Research Scientific Computing Center
(NERSC), a U.S. Department of Energy Office of Science User Facility located at Lawrence Berkeley National Laboratory, operated under
Contract No. DE-AC02-05CH11231.
We thank Hailong Wang for producing aerosol emissions input dataset based on the original input4MIPS data for the SSP5-8.5 all-forcing
experiment. We also thank Philip Cameron-Smith for producing GHGs and other forcing datasets. Figure 2 and Figure 3 are generated by
E3SM Diags (see https://e3sm.org/resources/tools/diagnostic-tools/e3sm-diagnostics) X. Zheng, J.-C. Golaz, and Q. Tang were supported
under the auspices of the U.S. Department of Energy by LLNL under contract DE-AC52-07NA27344.LLNL-JRNL-826361.





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





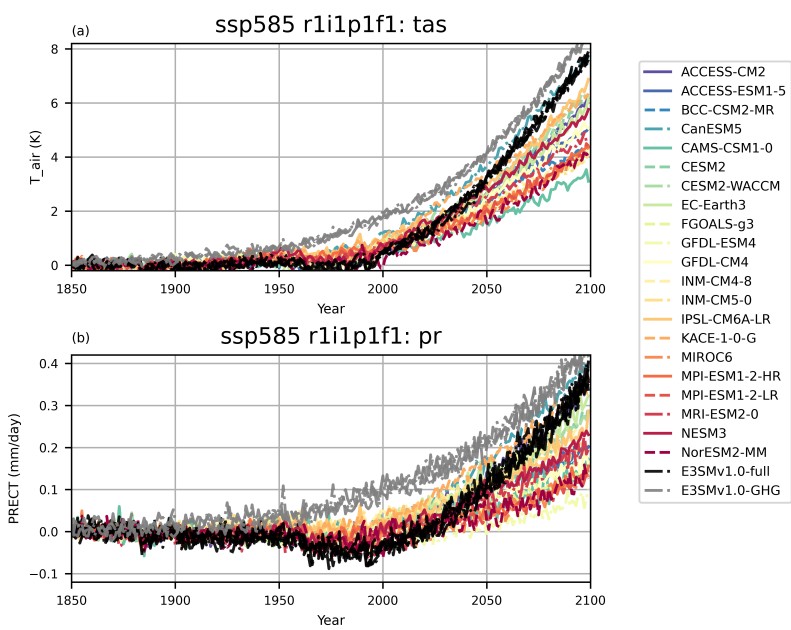

**Figure 1.** Time evolution of (a) annual global mean near-surface air temperature ($T_{air}$) anomalies, and (b) annual global mean surface precipitation rate anomalies with respect to 1850–1869 from E3SMv1.0 ensemble members (black lines) and CMIP6 models (colored lines) for the historical simulation and SSP5-8.5 all-forcing experiment. These three ensemble members of the E3SMv1.0 GHG-only experiments are denoted as gray dashed lines





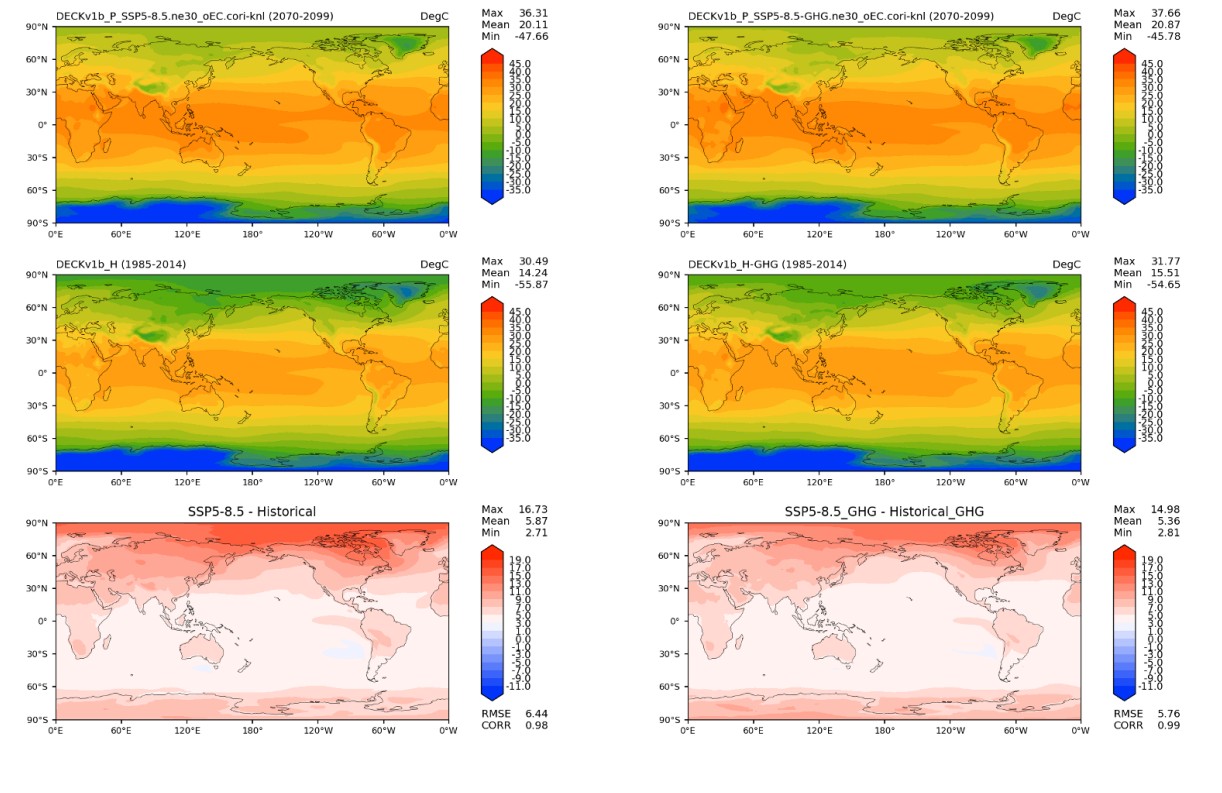

**Figure 2.** Left column: Annual mean $T_{air}$ ($^{o}C$) from (top) five SSP5.85 ensemble simulations (2070-2099), (mid) five historical ensemble simulations (1985-2014), and (bottom) the change between the time period of 2070 – 2099 and the period of 1985-2014. Right column: Annual mean $T_{air}$ ($^{o}C$) from (top) three SSP5.85-GHG ensemble simulations (2070-2099), (mid) three historical-GHG ensemble simulations (1985-2014), and (bottom) the change between the time period of 2070 – 2099 and the period of 1985-2014.





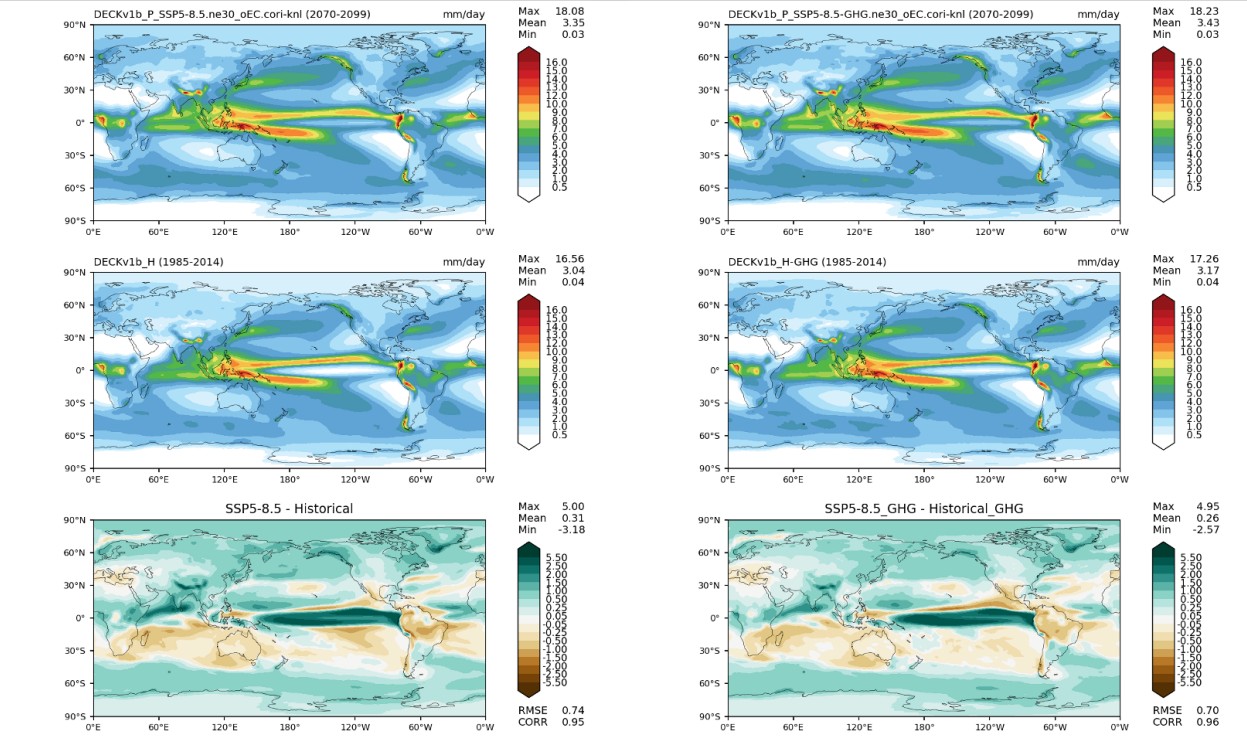

**Figure 3.** Left column: Annual mean precipitation rate (mm/day) from (top) five SSP5.85 ensemble simulations (2070-2099), (mid) five historical ensemble simulations (1985-2014), and (bottom) the change between the time period of 2070 – 2099 and the period of 1985-2014. Right column is same as the left column except for three ensemble members of the E3SMv1.0 GHG-only Experiment



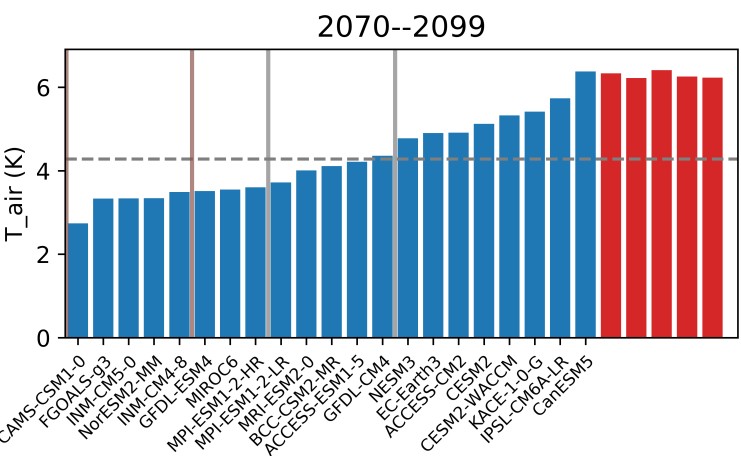

**Figure 4.** The changes of global mean ($T_{air}$) from year 1850-1869 to year 2070-2099 for CMIP6 models (blue bars) and E3SMv1.0 5 members from the SSP5-8.5 simulations. The five models between two vertical brown lines and these five models between gray vertical lines are models within $0 - 20th$ and $40 - 60th$ percentiles, respectively.





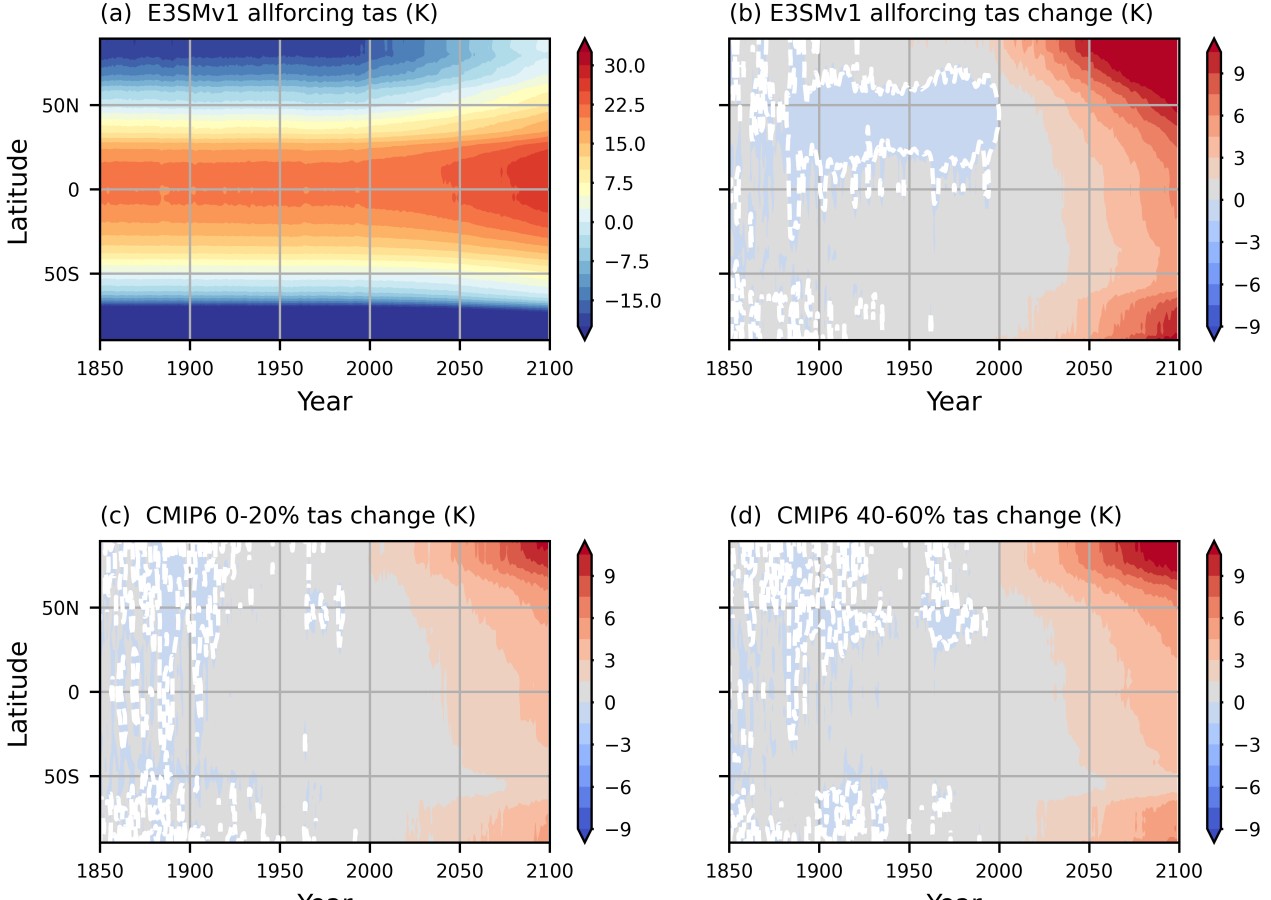

**Figure 5.** Time evolution of zonal mean (a) E3SMv1.0 annual ($T_{air}$) anomalies. Time evolution of the local changes in zonal mean $T_{air}$ with respect to 1850-1869 from the historical simulations and SSP5-8.5 simulations for (b) E3SMv1.0, (c) CMIP6 models within the $0-20th$ percentile range, and (d) CMIP6 models within the $40-60th$ percentile range based on Fig. 4



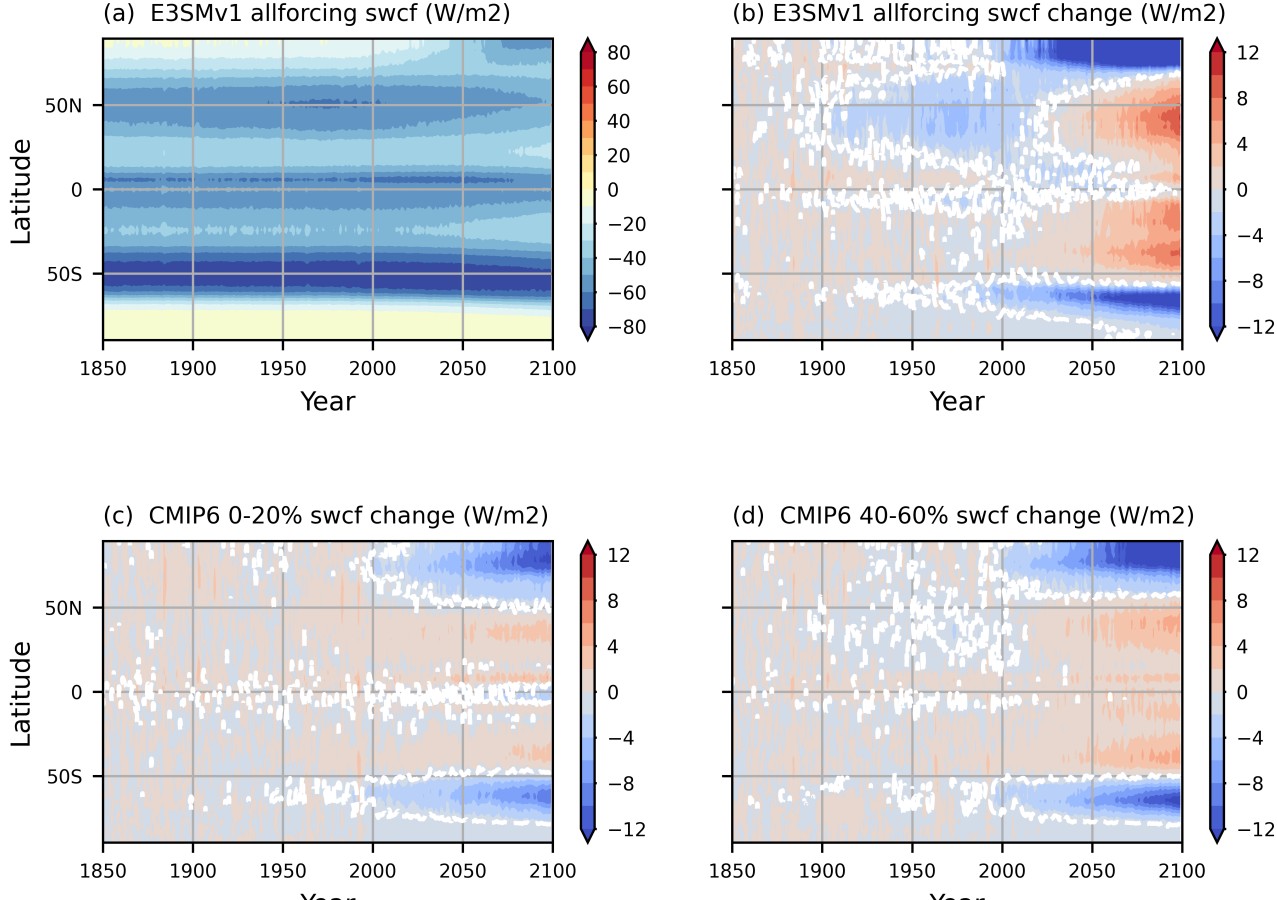

**Figure 6.** Time evolution of zonal mean (a) SWCRE for E3SMv1.0. Time evolution of the local changes in zonal mean SWCRE with respect to 1850-1869 from the historical simulations and SSP5-8.5 future climate simulations (b) E3SMv1.0, (c) CMIP6 models within the $0-20th$ percentiles , and (d) the $40-60th$ percentiles



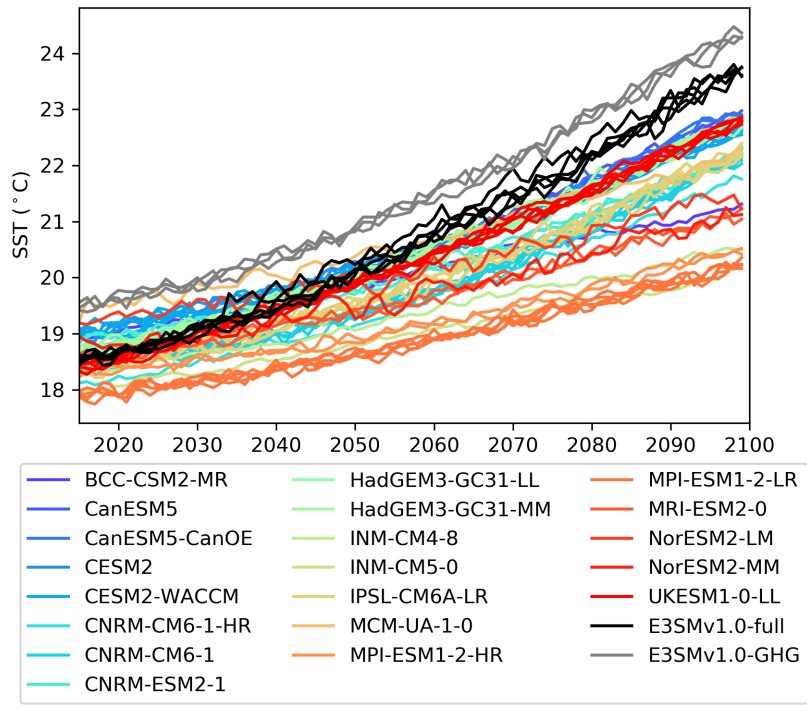

**Figure 7.** Time evolution of annual and global mean sea surface temperature (SST; ∘C) in E3SMv1.0 (black), E3SMv1.0 GHG-only (gray), and CMIP6 models (color) for the SSP5-8.5 experiment. Different ensemble members of the same model are denoted using the same color.



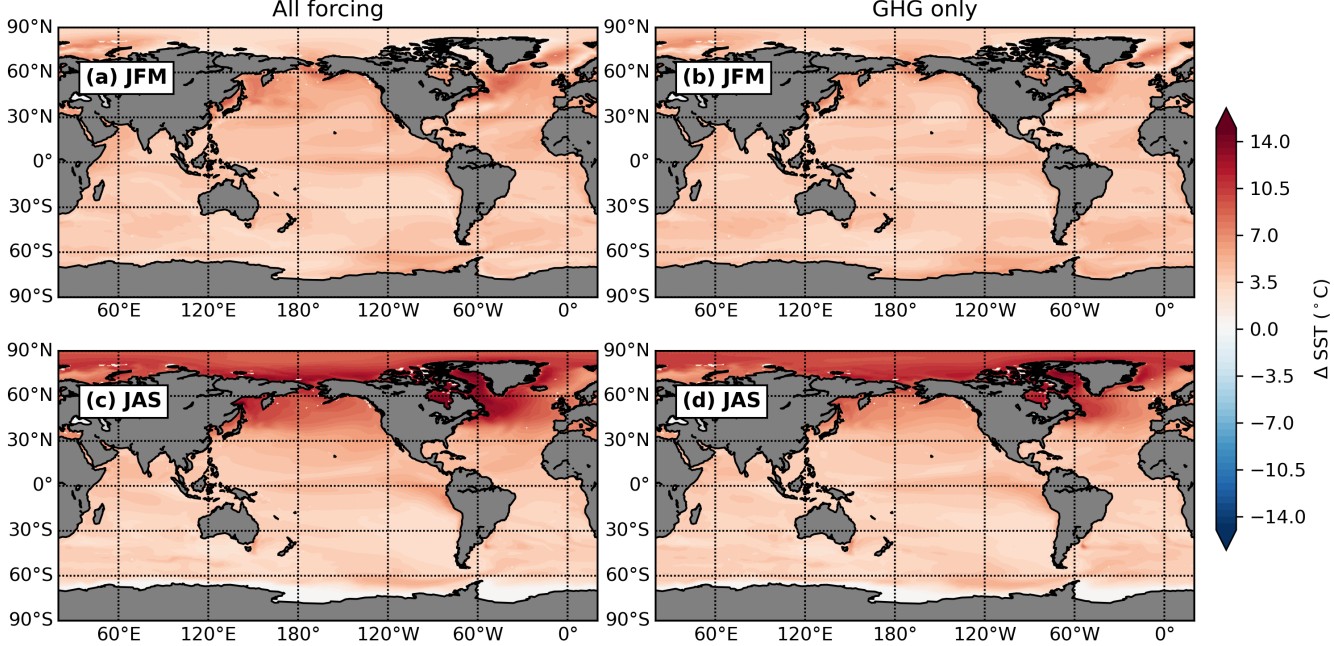

**Figure 8.** Changes of the ensemble averaged SST (°C) between the period of 2070-2099 and 1985-2014 in the boreal winter (a,b) and summer (c,d). (a) and (c) show the ensemble mean of the five E3SMv1.0 all-forcing simulations, and (b) and (d) show the ensemble mean of the three E3SMv1.0 GHG-only simulations.

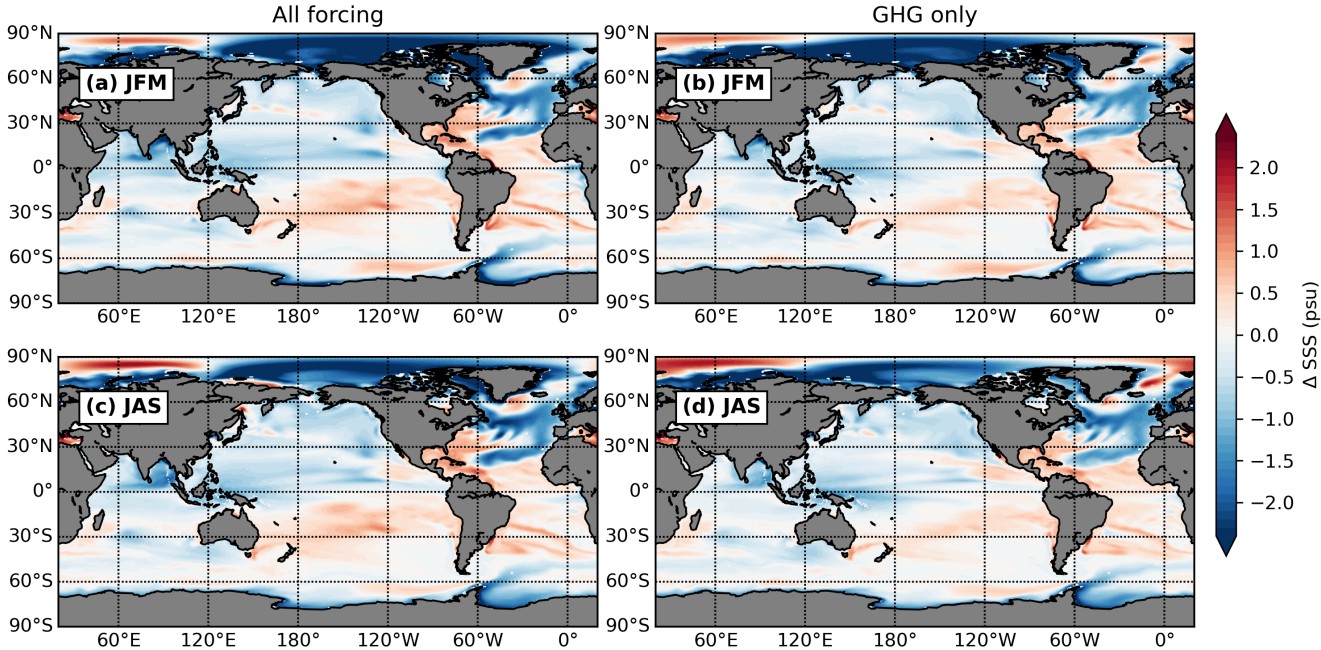

**Figure 9.** Same as Fig. 8, but for the sea surface salinity (SSS; psu).



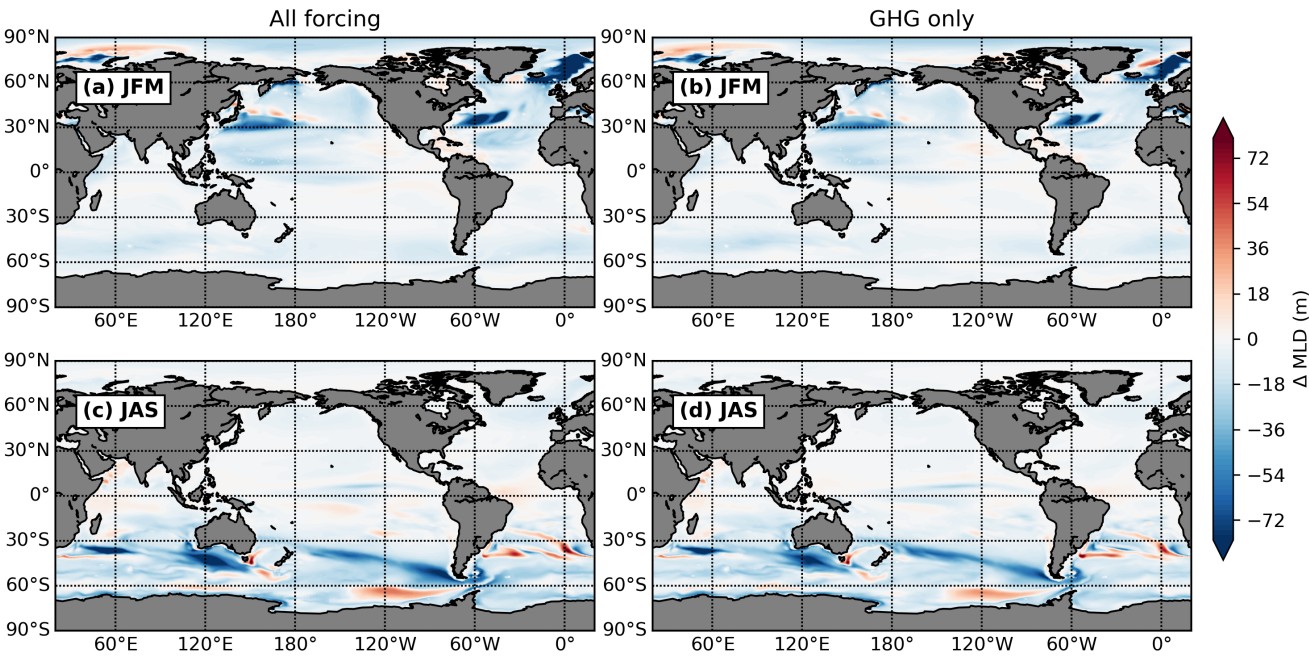

**Figure 10.** Same as Fig. 8, but for the mixed layer depth (MLD; m) based on a critical density threshold of 0.03 kg m$^{-3}$.





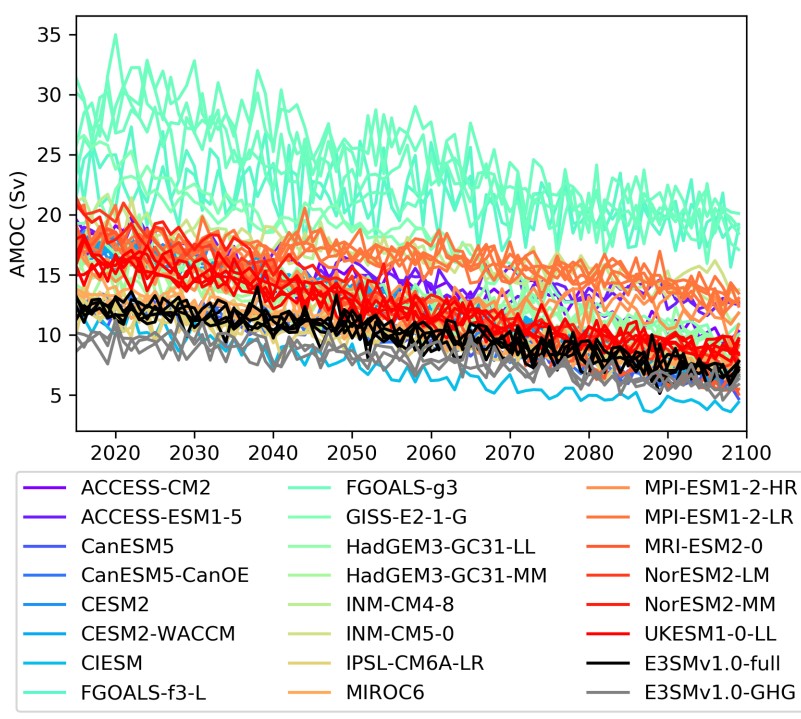

**Figure 11.** Same as Fig. 7, but for the Atlantic meridional overturning circulation (AMOC; 1 Sv=$1\times10^{6}$ m$^{3}$ s$^{-1}$) measured by the maximum streamfunction nearest to the RAPID Array latitude 26°N. Note that a slightly different set of CMIP6 models are shown due to data availability.



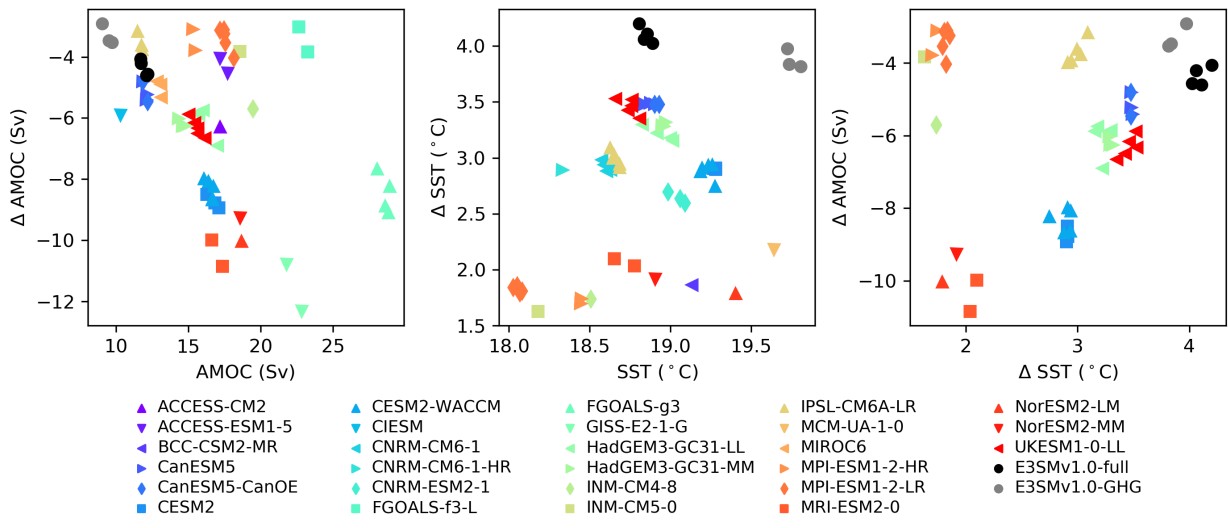

**Figure 12.** Scatter plot showing (a) the change in AMOC versus the reference AMOC, (b) the change in SST versus the reference SST, and (c) the change in AMOC versus the change in SST in the SSP-8.5 experiment. The reference AMOC and SST are the average over 2015-2034 and the changes are measured by the difference between the average over 2080-2099 and the reference.



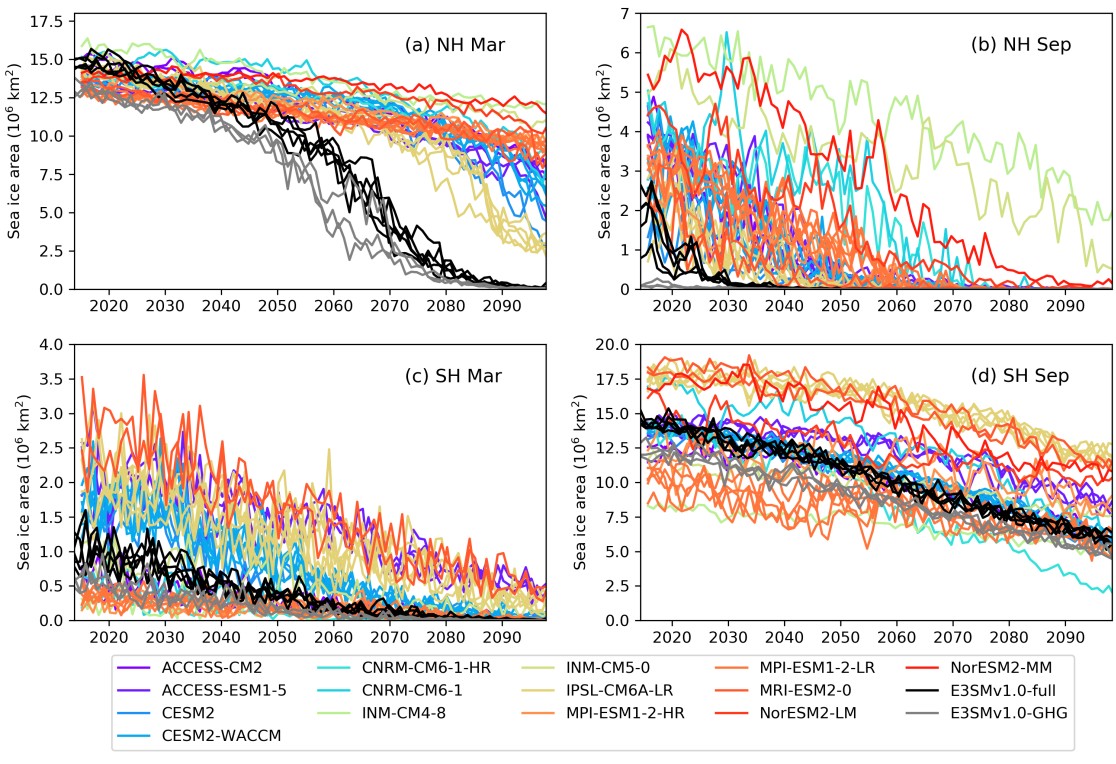

**Figure 13.** Same as Fig. 7, but for (a,b) the Northern Hemisphere and (c,d) Southern Hemisphere sea ice area ($10^6$ km$^2$) in (a,c) March and (b,d) September.





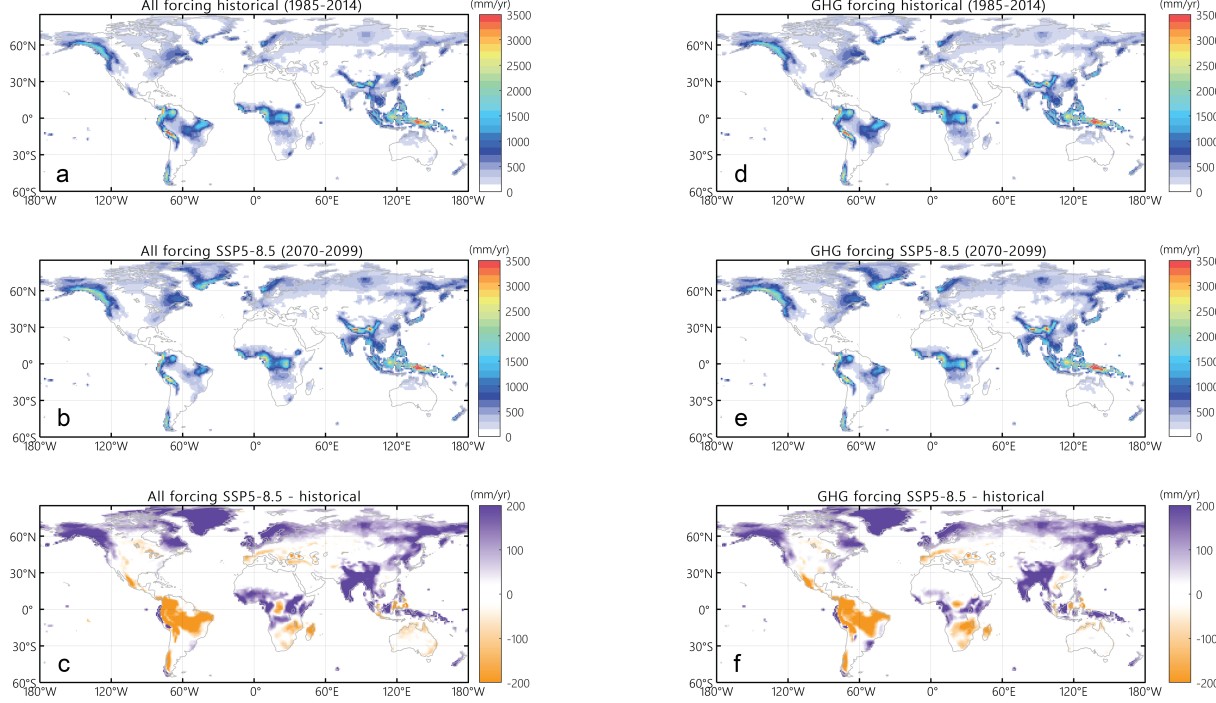

**Figure 14.** Left column: Annual mean runoff (mm/year) from (top) five historical ensemble simulations (1985-2014), (mid) five SSP5.85 ensemble simulations (2070-2099), and (bottom) the change between the time period of 2070 – 2099 and the period of 1985-2014. Right column is same as the left column except for three ensemble members of the E3SMv1.0 GHG-only Experiment





**Figure 15.** The time evolution of the zonal mean local (a) $T_{air}$ change (K), (b) SW cloud radiative forcing (W/m2) in E3SMv1.0 GHG-only simulations, and the simulated differences in (c) $T_{air}$ trend (K/year), (d) net cloud radiative forcing (W/m2), (e) Sulfate aerosol optical depth at 550 nm, and (f) total aerosol optical depth at 550 $nm$ between E3SMv1.0 all-forcing simulations and GHG-only simulations



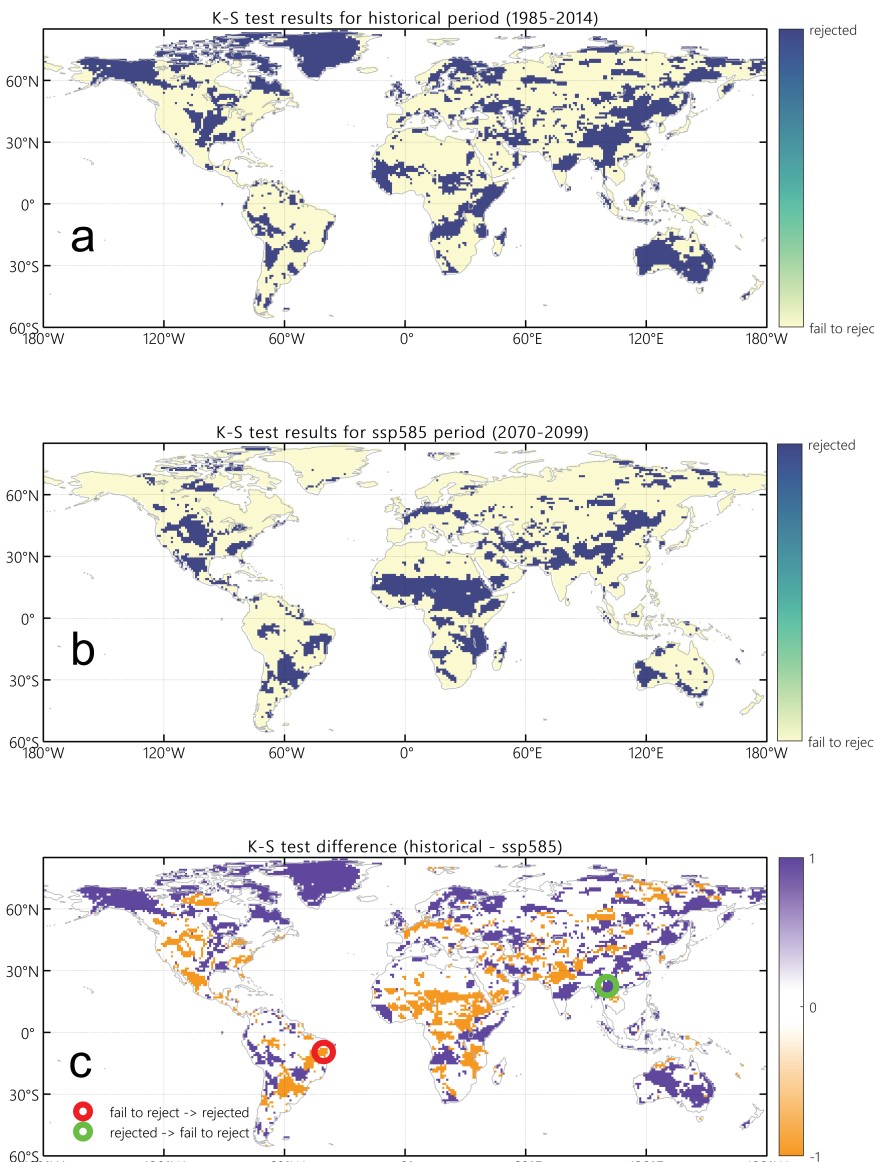

**Figure 16.** Two-sample K-S test results for mean annual runoff over (a) 30-year historical period (1985-2014), (b) 30-year ssp585 period (2070-2099), with "rejected" indicating the two samples are not from the same distribution and "fail to reject" indicating the two samples are from the same distribution. Circles on (c)the difference of the test results between the two periods indicate pixels selected for detailed time series and eCDFs demonstration in Fig. A1 and Fig. A2





Table A1: The list of the CMIP6 Models of which the historical experiments and ScenarioMIP experiments are adopted in this study through the Earth System Grid Federation (ESGF).

| Model | Model Center | Reference and ESGF |
|---|---|---|
| ACCESS-CM2 | CSIRO-ARCCSS (Australia) | Bi et al. (2020), https://doi.org/10.22033/ESGF/CMIP6.2285 |
| ACCESS-ESM1-5 | CSIRO (Australia) | Ziehn et al. (2020), https://doi.org/10.22033/ESGF/CMIP6.2291 |
| BCC-CSM2-MR | Beijing Climate Center (China) | https://doi.org/10.22033/ESGF/CMIP6.1732 |
| CAMS-CSM1-0 | Chinese Academy of Meteorological Sciences (China) | https://doi.org/10.22033/ESGF/CMIP6.11004 |
| CESM2 | National Center for Atmospheric Research (USA) | Danabasoglu et al. (2020), https://doi.org/10.22033/ESGF/CMIP6.2201 |
| CESM2-WACCM | National Center for Atmospheric Research (USA) | Gettelman et al. (2019), https://doi.org/10.22033/ESGF/CMIP6.10026 |
| CIESM | Tsinghua University (China) | Lin et al. (2020), http://doi.org/10.22033/ESGF/CMIP6.8863 |
| CNRM-CM6-1 | National Center for Meteorological Research (France) | Voldoire et al. (2019), https://doi.org/10.22033/ESGF/CMIP6.4224 |
| CNRM-ESM2-1 | National Center for Meteorological Research (France) | Séférian et al. (2019), http://doi.org/10.22033/ESGF/CMIP6.4226 |
| CanESM5 | Canadian Centre for Climate Modelling and Analysis (Canada) | https://doi.org/10.22033/ESGF/CMIP6.1317 |
| EC-Earth3 | EC-Earth Consortium (Europe) | Döscher et al. (2021), https://doi.org/10.22033/ESGF/CMIP6.4912 |
| FGOALS-f3-L | Institute of Atmospheric Physics (China) | https://doi.org/10.22033/ESGF/CMIP6.2046 |





| FGOALS-g3 | Institute of Atmospheric Physics (China) | https://doi.org/10.22033/ESGF/CMIP6.2056 |
|---|---|---|
| FIO-ESM-2-0 | First Institute of Oceanography (China) | https://doi.org/10.22033/ESGF/CMIP6.9214 |
| GFDL-CM4 | NOAA-Geophysical Fluid Dynamics Laboratory (USA) | https://doi.org/10.22033/ESGF/CMIP6.9242 |
| GFDL-ESM4 | NOAA-Geophysical Fluid Dynamics Laboratory (USA) | https://doi.org/10.22033/ESGF/CMIP6.1414 |
| GISS-E2-1 | Goddard Institute for Space Studies (USA) | Kelley et al. (2020), https://doi.org/10.22033/ESGF/CMIP6.7460 |
| HadGEM3-GC31 | Met Office (UK) | https://doi.org/10.22033/ESGF/CMIP6.10901 |
| INM-CM4-8 | Institute for Numerical Mathematic (Russia) | https://doi.org/10.22033/ESGF/CMIP6.12321 |
| INM-CM5-0 | Institute for Numerical Mathematic (Russia) | https://doi.org/10.22033/ESGF/CMIP6.12322 |
| IPSL-CM6A-LR | Institut Pierre-Simon Laplace (France) | https://doi.org/10.22033/ESGF/CMIP6.1532 |
| KACE-1-0-G | National Institute of Meteorological Sciences, Korea Meteorological Administration (South Korea) | https://doi.org/10.22033/ESGF/CMIP6.2241 |
| MCM-UA-1-0 | University of Arizona (USA) | http://doi.org/10.22033/ESGF/CMIP6.13901 |
| MIROC6 | JAMSTEC, NIES, AORI, U. of Tokyo (Japan) | https://doi.org/10.22033/ESGF/CMIP6.898 |
| MPI-ESM1-2 | Max Planck Institute for Meteorology (Germany) | https://doi.org/10.22033/ESGF/CMIP6.898 |
| MRI-ESM2-0 | Meteorological Research Institute (Japan) | https://doi.org/10.22033/ESGF/CMIP6.638 |





| NESM3 | Nanjing University of Information Science and Technology (China) | https://doi.org/10.22033/ESGF/CMIP6.2027 |
|---|---|---|
| NorESM2-MM | Norwegian Climate Center (Norway) | https://doi.org/10.22033/ESGF/CMIP6.608 |
| UKESM1 | Met Office and Natural Environment Research Council (UK) | Sellar et al. (2019), https://doi.org/10.22033/ESGF/CMIP6.6405 |



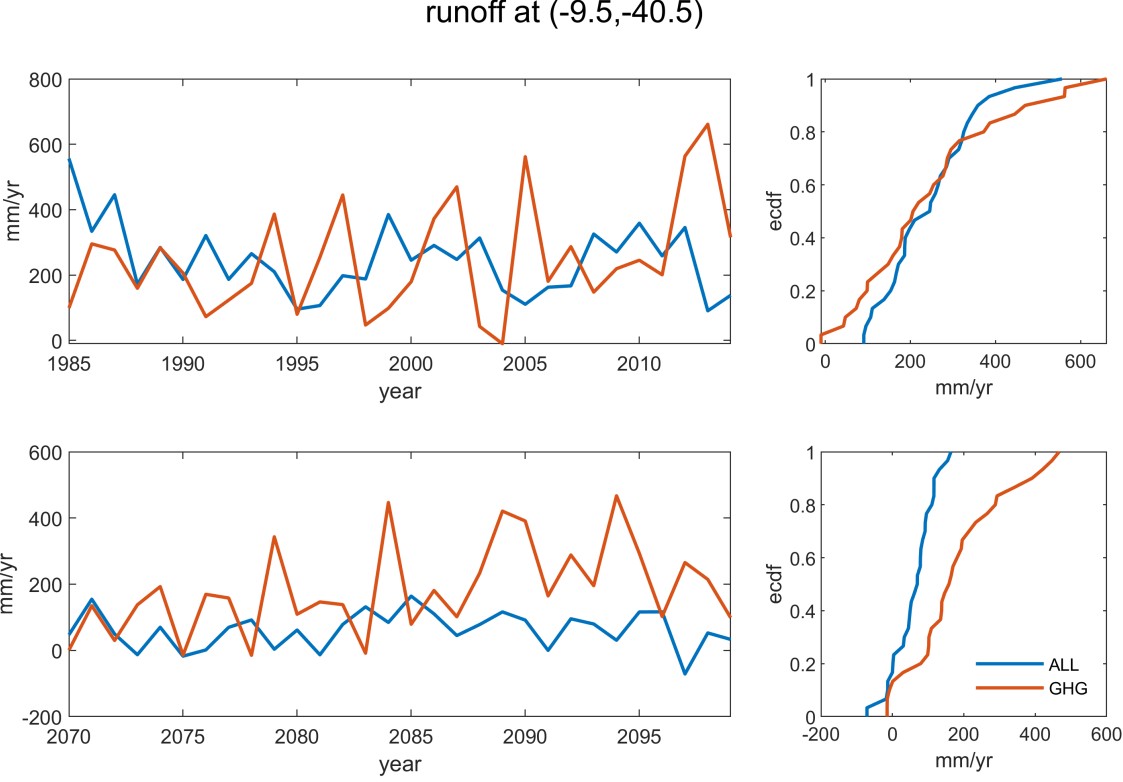

**Figure A1.** 30-year Mean annual runoff time series and eCDF from all forcing and GHG-only forcing simulations for historical period (upper panel), and SSP5-8.5 period(lower panel) at (9.5 ° S, 40.5 ° W) , where the K-S test was fail to reject (F) in the historical period and rejected (R) in the SSP5-8.5 period.



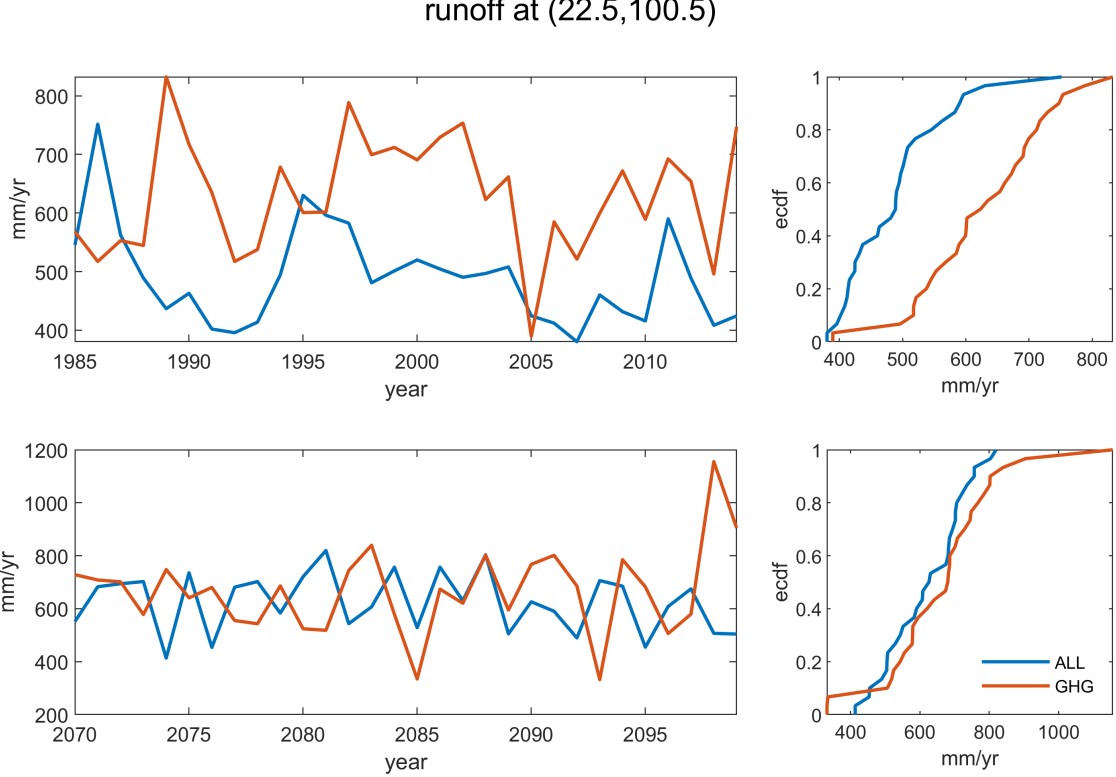

**Figure A2.** 30-year Mean annual runoff time series and ecdf from all forcing and GHG-only forcing simulations for historical period (upper panel), and SSP5-8.5 period(lower panel) at (22.5 ° N, 100.5 ° E) , where the K-S test was rejected (R) in the historical period and fail to reject (F) in the SSP5-8.5 period