# Peer review of "Description of historical and future projection simulations by the global coupled E3SMv1.0 model as used in CMIP6"

_Geoscientific Model Development, 2021_

## Referee Comment (RC2)

**Review of "Description of historical and future projection simulations by the global coupled E3SMv1.0 model as used in CMIP6" by Zheng et al., submitted to GMD**

**Major comments:**

The authors documented some future climate characteristics of E3SMv1.0 at the highest emission scenario, with a focus on regional responses. Moreover, the impact of anthropogenic aerosols on the warming was emphasized by comparing the SSP5-8.5 and SSP5-8.5-GHG simulations. This manuscript aims to describes the experiments and present the most notable features revealed in these experiments. It is found that the results are generally well presented. However, there is a lack of statistical significance when presenting the changes in future projections and the comparisons between SSP5-8.5 and SSP5-8.5-GHG. Therefore, I request minor revisions of the manuscript.

**Minor comments:**

Line 118−120: "CMIP6 models project an overall higher warming with a larger intermodal spread … compared to the corresponded CMIP5 future climate projects." Please cite related works.

Line 127: "E3SMv1.0 simulated global mean Tair anomalies" −> "The simulated global mean Tair anomalies in E3SMv1.0"; "demonstrates" −> "demonstrate"

Fig 3: It seems that there is an evident double-ITCZ problem in E3SMv1.0. Can you discuss a little about the impact of such bias on the projection?

Line 204: Please explain why January-February-March are used for boreal winter and July-August-September are used for boreal summer?

---

## Author Comment (AC1)

REVIEWER COMMENTS

*Response to Reviewer #1:*

***Thank you for the positive assessment and these constructive comments. The modifications that have been made to the text and figures to address these points are detailed below.***

Review #1 of "Description of historical and future projection simulations by the global coupled E3SMv1.0 model as used in CMIP6" by Zheng X. and co-authors.

The study documents the experimental setup and main characteristics of the historical and scenario simulations generated with the Earth System Model E3SMv1.0. The simulations follow the CMIP6 protocol under historical and the SSP5-8.5 scenario. The authors describe the main changes in global surface temperature and precipitation, ocean state, and runoff by 2100. The study also compares full-forcing scenario simulations with greenhouse-gas-only simulations, which allows the authors to assess the influence of aerosols in the slight cooling trend in the late 20th century.

The paper is important to document the experimental setup. It is clearly written and structured. I only have minor suggestions to extend and, in my view, improve the quality of the analysis presented, before the paper can be accepted for publication.

i) All variables may be compared over the historical period: while global surface temperature and precipitation, in Figure 1, are shown since 1850, all the other time series (Figures 7, 11, 13) start in 2014. Extending these time series back until 1850 too will help assessment of the historical part as well.

***All the time series have been extended back to 1850 to include the results of the historical simulations (Fig. 7, 13). The discussions of these figures in the text are also modified accordingly.***

ii) The paper will benefit if some other variables are also included to assess dynamical changes in the climate: for example, changes (maps) in winter vs. summer precipitation, atmospheric jets (via zonal wind at 250 hPa), storm tracks, ENSO variability (spectrum over reference periods), sea ice volume, or the spatial pattern of the AMOC (in addition to the time series).

***We agree with the reviewer. We have added figures and descriptions of the changes in zonal wind at 250 hPa (Fig. 4) and DJF vs JJA precipitation (Fig. A1, A2) to briefly assess dynamical changes in the climate (Lines 174-179). Furthermore, we have added a figure showing the spatial pattern of the AMOC and the changes under SSP5-8.5 forcing scenario by comparing the ensemble mean AMOC over the 30-year period of 1985-2014 and 2070-2099 (Fig. 12). While we acknowledge the importance of analyses on climate variability such as ENSO variability, our analysis is limited by the relatively small number of ensemble members (five) in our SSP5-8.5 simulations and relatively short period (2015-2099) in each simulation under transient forcing. We are working on another paper that focuses more on the tropical***

*Pacific under the SSP5-8.5 forcing scenario, specifically the tropical instability waves and their relationship with ENSO. In that analysis we noted a stronger amplitude of the Nino SST indices and a slight shift of ENSO towards the central tropical Pacific under SSP5-8.5 forcing scenario. But it is hard to detect a robust change in ENSO without a large set of ensemble simulations. We have noted this limitation of our analysis here in the text (Lines 463-465).*

iii) Whenever possible, I suggest including observed time series as a reference.

*Following this suggestion, observed time series have been added to the time series figures of $T_{air}$, Precipitation, SST, AMOC and sea ice area (Fig. 1, Fig. 7, and Fig. 13). Due to the limited period of the available precipitation observation (1979-2017), we plot the time series of the observed precipitation change relative to 1979 instead of precipitation anomalies with respect to 1850–1869.*

**Minor points:**

L7: You can give the value of the model's climate sensitivity here too.

*Done (Lines 34-35).*

L8: Can you elaborate on how changes in runoff respond to precipitation changes.

*We have added the correlation coefficients between precipitation and runoff in terms of spatial distribution and temporal variation (Lines 281-286).*

L9-11: I suggest talking first about all oceanic changes (mixed layer depth, AMOC) and then about sea ice.

*We agree with you. The content has been rearranged accordingly (Lines 10-12).*

L11: Can you give the AMOC weak strength as a reference?

*Yes, we have added the AMOC strength. (Line 11).*

L16: Certain regions: Which ones in particular?

*We reworded it to "… over certain regions, e.g., southern North America, southern Africa,*

*central Africa, and eastern Asia." (Lines 17-18).*

L33: Is it known why the climate sensitivity is so high in this model?

*Zelinka et al. (2020) found that the too-strong positive cloud feedback is the primary reason for the overly high climate sensitivity. We have added their finding in the manuscript. (Lines 36-37)*

L128: at the lower end: this is hard to see in the Figure because it is clipped at 0. I suggest extending the vertical axis.

*The vertical axis has been updated (Fig. 1).*

L136: Is there a value for the correlation?

*We have added the correlation coefficient: "The median/mean values of the Pearson correlation coefficients for Tair and precipitation anomalies from the CMIP6 models and the E3SMv1.0 simulations are higher than 0.96." (Lines 141-143).*

L140-143: The temperature pattern can briefly be described too, as done for precipitation.

*A brief description of the annual mean temperature pattern has been added before the precipitation pattern as follows: "The global pattern of Tair change between 2070-2099 and 1985-2014 reveals a polar amplification of surface warming and inter-hemispheric asymmetric warming. The strongest surface warming occurs over the Arctic with a magnitude > 15ºC. Along the same latitude bands, the surface warming over the continents is generally higher than oceans." (Lines 150-156).*

L143: I would also say the ITCZ seems to narrow over the central and eastern Pacific while it seems to shift northward over the Indian and Atlantic (increasing their precipitation over their monsoon regions).

*This part of the text has been revised accordingly "Furthermore, the ITCZ over the central and eastern Pacific Ocean becomes narrower, whereas it shifts northward over the Indian and Atlantic Oceans with more precipitation over the monsoon regions." (Lines 159-160).*

L154-157: Because it is directly related to the Tair temperature series, the description of Figure 4 can come before the description of the maps.

*Done (Lines 146-149).*

L157: I suggest a paragraph break before "Besides".

*The original paragraph has been separated before "Besides" (Line 180).*

L161: It is not entirely clear why the mention of the Golaz et al. (2019) paper on SST anomalies is relevant here. Can you clarify it?

*We agree that the SST anomalies in Golaz et al. (2019) is not relevant. This sentence has been deleted.*

L187: A substantial difference is not very precise. Can you give a value?

*Based on Fig. R1 (not shown in the manuscript), we revised the description to "Compared with CMIP6 models with medium warming, while the E3SMv1.0 simulated negative SWCRE change in the Arctic is stronger than -12 W/m2 after 2050, the positive SWCRE change in both Northern and Southern Hemispheric low- and mid-latitudes increases after 2050 and the difference exceeds 8 W/m2 by 2100 (Fig. 6)." (Lines 206-209)*

[Figure]

E3SMv1 allforcing - CMIP6 40-60% swcf change (W/m2)

Figure R1. The difference in the time evolution of the local changes in zonal mean SWCRE with respect to 1850-1869 from the historical simulations and SSP5-8.5 simulations between the E3SMv1.0 and CMIP6 models within the 40-60th percentile range based on Figure 1(c).

L192: Give approximate periods when there is asymmetric cooling and warming.

*The sentence has been revised to "Throughout the historical period and the future climate projection period, E3SMv1.0 produces an inter-hemispheric asymmetric cooling between 1900 and 2000, followed by an inter-hemispheric asymmetric warming until the end of the 21st century, both of which are closely linked to the cloud responses, especially in the Northern Hemisphere." (Lines 211-214)*

L215: Can you give the mean values of the simulated and observed AMOC that are here compared?

*Added "The mean AMOC simulated in the E3SMv1.0 historical ensemble (~11 Sv) is weaker than the observed mean (16.9 Sv) and the ensemble mean of CMIP6 models (Weijer et al., 2020)." (Lines 231-232)*

L234: contributes to.

*Corrected.*

L240: Runoff also decreases over the Amazon basin (assuming this is not included in Central America).

***This sentence has been revised to "with decreased runoff in the Mediterranean region, southern Africa, southern North America, northern South America, Australia, and increased runoff in high latitudes of the Northern Hemisphere, central Africa, as well as southern to eastern Asia." (Lines 279-281).***

L245: The correlation coefficient between runoff and precipitation changes is needed.

***We have provided the Pearson correlation coefficients between the mean annual precipitation and mean annual runoff over the historical period (1985-2014) and future projection period (2070-2099) respectively (Fig. R2). The values (0.9 and 0.89) suggest a high similarity in spatial pattern between the two variables. The annual precipitation and runoff anomaly time series across the entire simulation period (1850-2099) over land has a Pearson correlation coefficient as high as 0.99 (Fig. R3), indicating a strong correlation in terms of variation between precipitation and runoff. We have included these values in section 3.13 "Land Climatology". (Lines 282-286)***

[Figure]

Figure R2. Spatial correlation coefficient between historical mean annual precipitation (a) and mean annual runoff (b), as well as correlation coefficient between future mean annual precipitation (c) and mean annual runoff (d).

[Figure]

Figure R3. Scatter plot between spatially averaged annual mean annual precipitation anomaly and runoff anomaly over the entire historical and future projection period (1850-2099). Each circle represents one year.

L244: It is not clear what is meant by "the position of E3SMv1.0".

*We reworded this sentence to "Given that the spatial distribution and bias of the runoff are highly consistent with those in precipitation for E3SMv1.0, it is fair to presume that the position of E3SMv1.0 simulated global runoff in the CMIP6 ensemble spread is also similar to the global mean precipitation in the ensemble spread (Fig. 1b)." (Lines 286-288).*

L253: Contribute to

*Corrected.*

L255: Please, clarify "high-level features"

*The sentence has been revised to "Beyond these general features of the changes in surface temperatures, sea-ice, and the AMOC, the following subsections focus on the differences between the historical and SSP5-8.5 experiments (i.e., the all-forcing simulations) and the corresponding GHG-only experiments …" (Lines 297-300).*

L259: "In the absence of all other external forcing,"

*Corrected.*

L269: This warming slowdown is hard to see. In Figure 1, the Tair trend (as the first derivative of the curve) in the full-forcing ensemble progressively increases until it matches the trend in the GHG-only runs by 2100. In Figure 2 the smaller polar amplification in the GHG-only runs compared to the full-forcing ones is because the mean temperature of the reference period is initially different. In Figures 5 and 15a any reduction in the warming rate is also difficult to see because the color scale saturates above 9K. And in Figure 15c, the difference in trend by 2100 are relatively small (0-0.5 K in 30 year) and of different sign in the NH and SH, which compensate each other globally (so the similar trends in Fig. 1a). I would propose a simpler interpretation of these figures, with the full-forcing runs having the GHG-related warming delayed by the cooling effect of aerosols, which, once removed, leads to a transient faster warming in the first half of the 21$^{st}$ century.

***We conducted a two-sided t-test with the null hypothesis that the all-forcing runs and GHG-only runs give identical ensemble mean for the $T_{air}$ trend. The result shows that the cooling trend before 2000 and the warming trend of the all-forcing experiment relative to the GHG-only experiment between 2000 and 2050 are statistically significant; however, the warming trend during the last 50 years of simulation is not statistically significant, which supports the reviewer's argument. As a result, the description in section 3.2.1, conclusion, and abstract have been updated to reflect the revised interpretation.***

L297: "increases more significantly" no statistical test has been included to prove this.

***A two-sided t-test with the null hypothesis that the GHG-only experiment and all-forcing experiment give identical ensemble mean was conducted. Regions of significant changes are highlighted using stippling dots in these maps. We also revised the corresponding discussions of these figures in the text (Lines 344-353).***

L303: "reducing deep convection" This is partially shown in the Figure 10.

***Thanks for the comment. Note that Figure 10 (now Figure 11 with modifications) shows the changes of MLD comparing the SSP5-8.5 and historical simulations. But this note is based on the fact that the MLD in the high-latitude North Atlantic is shallower in the GHG-only experiment than in the all-forcing experiment (not shown).***

L310: Changes in SST are not only necessary driven by ocean dynamics. There can be thermodynamic processes as well, for example, related to increases energy flux through the surface.

***We agree that changes in SST are not necessarily driven by ocean dynamics and therefore not a good indicator of changes in ocean dynamics. We have rewritten the discussion of the sea ice area time series here (Lines 358-372).***

Section 3.2.3: I would propose extending the Kolmogorov-Smirnov test to the other variables compared between the full-forcing and GHG-only runs to highlight were anomalies are statistically significant. Also, I would add the information in Figure 16 to Figure 14, for example as a stippling masking non-significant anomalies, which is a standard way of including

significance test on plots. In the current format, comparing Figure 14 and 16 by eye is difficult. The statistical test can compare the reference period vs. the late-21st century period for the full-forcing plots, and the full-forcing vs. GHG-only late-21st century differences for the GHG-only plots.

*In this manuscript we use different statistical methods to compare all-forcing experiments against GHG-only experiments. For variables in atmospheric and ocean components such as Tair and SST, we have already conducted the t-tests to identify significant differences in annual mean between the full-forcing and GHG-only experiments. For the runoff, we provide a different angle by investigating distribution using the K-S test. We agree with the reviewer that adding more tests to the analyses would enrich the results and discussion. But by leveraging the extra values added by the new tests versus the focus of the manuscript, we decide to keep one statistical test per variable to keep section 3.2 concise.*

*Regarding merging Fig. 14 and Fig. 16 together, we have tried the method suggested by the reviewer. However, the outcome is not as good as we expected. The spatial patterns showed in Figure 16 cannot be clearly identified/compared when overlaying to Fig. 14. Therefore, we made another attempt by simply stitching Fig. 14 and Fig. 16 together (see. Fig. R4). It is easier for intercomparison but the plot in each panel is much smaller in this form. In addition, we are unable to show the two highlighted point in Fig. 16 on the new plot given the small size. With all things considered, we respectfully decided to keep the original form.*

[Figure]

Figure R4. Combined Figure 14 and Figure 16

Figure 4 may be merged with Figure 1 (if new figures with new variables are to be included).

*The original Fig. 4 has been merged to Fig. 1 as Fig. 1c now. Thank you for the suggestion.*

Figure 12 is barely discussed. It could be removed or moved to the supplement. Also, why the reference periods are different, for example, compared to Figure 2?

*We feel this is an interesting figure showing the relation between the changes in SST and changes in AMOC. We have rearranged the order of the figures and corresponding discussions for clarity. The time series of AMOC has been merged with the time series of SST and this figure is now presented directly after the time series of SST and AMOC. We have also slightly extended the discussion of this figure (Lines 236-245).*

---

## Author Comment (AC2)

*Response to Reviewer #2:*

*We sincerely thank the reviewer for taking time to review our manuscript and offering constructive comments and suggestions. The modifications and answers that have been made to address the reviewer's concerns are listed below.*

Review of "Description of historical and future projection simulations by the global coupled E3SMv1.0 model as used in CMIP6" by Zheng et al., submitted to GMD

Major comments:
The authors documented some future climate characteristics of E3SMv1.0 at the highest emission scenario, with a focus on regional responses. Moreover, the impact of anthropogenic aerosols on the warming was emphasized by comparing the SSP5-8.5 and SSP5-8.5-GHG simulations. This manuscript aims to describes the experiments and present the most notable features revealed in these experiments. It is found that the results are generally well presented. However, there is a lack of statistical significance when presenting the changes in future projections and the comparisons between SSP5-8.5 and SSP5-8.5-GHG. Therefore, I request minor revisions of the manuscript.

*To address the reviewer's comment about the statistical significance of the comparisons, a two-sided t-test with the null hypothesis that the GHG-only experiment and all-forcing experiment give identical ensemble mean was conducted for the mean SST, SSS and MLD. Regions of significant changes are highlighted using stippling dots in the maps. We also revised the corresponding discussions of these figures in the text (Lines 344-353). In addition, the same t-test has been conducted for the simulated difference in Tair trend, net cloud radiative forcing, and aerosol optical depths between E3SMv1.0 all-forcing simulations and GHG-only simulations (Fig. 15). We also revised the relevant content (Lines 315-317).*

Minor comments:
Line 118−120: "CMIP6 models project an overall higher warming with a larger intermodal spread … compared to the corresponded CMIP5 future climate projects." Please cite related works.
*We cited three previous works as references "(e.g., Meehl et al., 2020; Brunner et al., 2020; Tebaldi et al., 2021)" (Lines 124-125).*

Line 127: "E3SMv1.0 simulated global mean Tair anomalies" −> "The simulated global mean Tair anomalies in E3SMv1.0"; "demonstrates" −> "demonstrate"

*Corrected. Thank you.*

Fig 3: It seems that there is an evident double-ITCZ problem in E3SMv1.0. Can you discuss a little about the impact of such bias on the projection?

*We added a brief discussion about the double-ITCZ bias and its potential impact with a couple of references: "As shown in Fig. 3, E3SMv1.0 has the double-ITCZ bias that is persistent in generations of CMIP models (Tian and Dong, 2020). The double-ITCZ bias is found to have a large impact on the projection of precipitation and tropical climate change. Specifically, the projected precipitation change tends to be proportional to the precipitation bias in the double-ITCZ regions (Brown et al., 2015; Zhou and Xie, 2015; Samanta et al., 2019)." (Lines 164-167)*

Line 204: Please explain why January-February-March are used for boreal winter and July-August-September are used for boreal summer?

*The ocean has a larger thermal inertia than the atmosphere so there is a delay in the seasons in the ocean as compared to the atmosphere. It is conventional to use January-February-March for boreal winter and July-August-September for boreal summer to describe the seasonal variation of ocean variables. This explanation has been added in the manuscript (Lines 246-248).*